



# Impact of the quality of hydrological forecasts on the management and revenue of hydroelectric reservoirs - a conceptual approach

Manon Cassagnole[1], Maria-Helena Ramos[1], Ioanna Zalachori[1,2], Guillaume Thirel[1], Rémy Garçon[3], Joël Gailhard[3], and Thomas Ouillon[4]

[1]Université Paris-Saclay, INRAE, UR HYCAR, 1 Rue Pierre-Gilles de Gennes, 92160 Antony, France
[2]Now at TERNA ENERGY, Hydroelectric Projects Department, Athens, Greece
[3]EDF-DTG, Electricité de France, Direction Technique de Grenoble, France
[4]EDF-Lab, Electricité de France, Paris Saclay, France

**Correspondence:** M.H. Ramos (maria-helena.ramos@inrae.fr)

**Abstract.** The improvement of a forecasting system and the continuous evaluation of its quality are recurrent steps in operational practice. However, the systematic evaluation of forecast value or usefulness for better decision-making is less frequent, even if it is also essential to guide strategic planning and investments. In the hydropower sector, several operational systems use medium-range hydrometeorological forecasts (up to 7-10 days ahead) and energy price predictions as input to models

that optimize hydropower production. The operation of hydropower systems, including the management of water stored in reservoirs, is thus partially impacted by weather and hydrological conditions. Forecast value can be quantified by the economic gains obtained with the optimization of operations informed by the forecasts. In order to assess how much improving the quality of hydrometeorological forecasts will also improve their economic value, it is also essential to understand how the system and its optimization model are sensitive to sequences of input forecasts of different quality. This paper investigates

the impact of 7-day streamflow forecasts of different quality on the management of hydroelectric reservoirs and the economic gains generated from a linear programming optimization model. The study is based on a conceptual approach, where inflows to 10 reservoirs in France are synthetically generated over a 4-year period to obtain forecasts of different quality in terms of accuracy and reliability. Relationships between forecast quality and economic value (hydropower revenue) show that forecasts with a recurrent positive bias (overestimation) and low accuracy generate the highest economic losses, when compared to the

reference management system where forecasts are equal to observed inflows. The smallest losses are observed for forecast systems with under-dispersion reliability bias, while forecast systems with negative bias (underestimation) show intermediate losses. Overall, the losses represent approximately 3% to 1% (in M€) of the revenue over the study period. Besides revenue, the quality of the forecasts also impacts spillage, stock evolution, production hours and production rates, with systematic over- and under-estimations being able to generate some extreme reservoir management situations.

## 1 Introduction

According to the 2018 report of the International Hydropower Association (IHA, 2018), the worldwide total generating capacity of hydropower plants is more than 1200 GW, making hydropower the world's leading renewable energy source. The share





of global renewable energy production was 25.6% in 2018, of which 15.9% came from hydroelectric production. In France, hydropower is expected to play a central role in meeting the flexibility needs of the evolving electricity system under the

clean energy transition. The country has 25.5 GW of installed hydropower capacity, which makes it the third largest European producer of hydroelectricity in Europe (IHA, 2018). Among the existing hydropower plants, more than 80% are operated by Électricité de France (EDF). EDF develops in-house forecasting systems to forecast river discharges and reservoir inflows in catchments of sizes ranging from a few tens to thousands of squared kilometres. Most catchments are located in mountainous areas and are governed by different hydrological regimes, from glacio-nival to more pluvial-dominant runoff regimes. Fore-

casting systems allow anticipating hydro-meteorological conditions from hours to days and months ahead. Over the years, investments have been made to develop deterministic and probabilistic forecast products which meet the requirements of being reliable and accurate. Usefulness is ensured by constant interaction with users. However, several questions remain: are investments in forecast quality rewarding in terms of economic benefits and improved hydropower production management? How does forecast quality impact reservoir management and hydropower revenues?

Operating a hydroelectric reservoir involves deciding when it is more interesting to produce energy (i.e. use the water stored in the reservoir by releasing it through the turbines of the power plant), and when it is more interesting to store the water in the reservoir to use it later, when demand (and electricity prices) are higher. Management decisions are also affected by other roles the reservoir may have within an integrated river basin management (e.g. irrigation for agriculture, flood control, drought relief) as well as by management constraints (e.g. reservoir capacity, production capacity), which are specific to each reservoir. To

help reservoir management decision-making, tools exist that model the management problem and help to find the most optimal sequence of releases in order to fulfill the management objectives. In the literature, there are several optimization algorithms used to manage hydroelectric reservoirs. Dobson et al. (2019), Rani and Moreira (2010), Ahmad et al. (2014), Celeste and Billib (2009) and Labadie (2004) carried out extensive reviews of the most common optimization methods. Three main classes of optimization algorithms that are efficient for optimizing reservoir management are: (1) linear programming (Arsenault and

Côté, 2019; Yoo, 2009; Barros et al., 2003), (2) dynamic programming (Bellman, 1957) and its variants, deterministic dynamic programming (DDP) (Haguma and Leconte, 2018; Ming et al., 2017; Yuan et al., 2016), stochastic dynamic programming (SDP) (Wu et al., 2018; Yuan et al., 2016; Celeste and Billib, 2009; Tejada-Guibert et al., 1995), sampling stochastic dynamic programming (SSDP) (Haguma and Leconte, 2018; Faber and Stedinger, 2001; Kelman et al., 1990) and stochastic dual dynamic programming (SDDP) (Macian-Sorribes et al., 2017; Tilmant et al., 2011, 2008; Tilmant and Kelman, 2007; Pereira and Pinto, 1991), and (3) heuristic programming (Macian-Sorribes and Pulido-Velazquez, 2017; Ahmed and Sarma, 2005).

The choice among these algorithms depends on many factors, such as the stakes and objectives to address, as well as the configuration of the system and the data available to parametrize and run the model.

Among the data used in reservoir management and operation, the water inflows to the reservoir, characterized by their time variability, are crucial, either if they are observed hydrologic flows, or forecasts. Streamflow forecasts provide short- to long-

term information on the possible scenarios of inflows and, consequently, affect the decisions to be made on releases and storage. The effectiveness of an optimization model may thus depend on how good these forecasts are. Murphy (1993) lists three main aspects that define if a forecast is good: consistency, quality and value. Forecast consistency relates to the correspondence



between the forecast and the forecaster's best judgement, and depends on the forecaster base knowledge. Forecast quality relates to how close the forecast values (or the forecast probabilities) are to what actually happened. Forecast value relates to
the degree to which the forecast helps in a decision-making process and contributes to realize economic, or other, benefit.

Forecast quality is often characterized by attributes, such as reliability, sharpness, bias and accuracy. It is often assessed with numeric or graphic scores, and independently of forecast value. When forecasts are affected by errors and display biases or inaccuracies, they can be improved by applying statistical corrections, also called post-processing techniques, to the biased forecasts. Post-processing is widely discussed in the literature (Ma et al., 2016; Crochemore et al., 2016; Thiboult and Anctil,
2015; Pagano et al., 2014; Verkade et al., 2013; Trinh et al., 2013; Gneiting et al., 2007; Fortin et al., 2006; Gneiting et al., 2005), for deterministic and probabilistic (or ensemble-based) forecasts. It is also widely demonstrated that multi-scenario, ensemble forecasts provide forecasts of better quality and enhanced potential usefulness when compared to single-value deterministic forecasts, even when the mean of all ensemble members is used (Fan et al., 2015; Velázquez et al., 2011; Boucher et al., 2011; Roulin, 2007).

While the analysis of forecast quality receives large attention, with numerous scores developed to quantify quality gains when improving a forecasting system, the evaluation of forecast value remains a challenge. The value of a forecast represents the benefits realized through the use of the forecast in decision-making. It is therefore necessary to acquire knowledge on how decisions are made when informed by forecasts. In the context of hydroelectric reservoir management, the value of a forecast is often assessed by the performance and benefits obtained from an optimal management, when management objectives
are satisfied and constraints are respected (storage capacity, environmental constraints). It can be expressed: (1) in terms of economic revenues, often associated with a monetary unit (Arsenault and Côté, 2019; Tilmant et al., 2014; Alemu et al., 2011; Faber and Stedinger, 2001), and (2) in terms of utility, often associated with a production unit (Côté and Leconte, 2016; Desreumaux et al., 2014; Boucher et al., 2012; Tang et al., 2010).

The analysis of the relationship between the quality of hydrometeorological forecasts and their economic value in the hy-
droelectric sector is more frequent in the context of seasonal hydropower reservoir management. For example, Hamlet et al. (2002) show that the benefits generated by the use of seasonal forecasting for the management of a water reservoir used for irrigation, hydropower production, navigation, flood protection and tourism can reach 153 million dollars per year. Boucher et al. (2012) studied the link between forecast quality and value at shorter, days ahead, lead times. The authors reforecast a flood event that occurred in the Gatineau river basin in Canada due to consecutive rainfall events and evaluate the management
of the Baskatong hydropower reservoir under different inflow forecast scenarios. They show that the use of deterministic or raw (without bias correction) ensemble streamflow forecasts does not affect the forecast management value. However, the use of a post-processor to correct ensemble forecast biases lead to a better reservoir management.

In order to better understand the relationship between the quality of hydrometeorological forecasts and their value through the management of hydroelectric reservoirs, some studies have created synthetic, quality-controlled hydrological forecasts.
The use of synthetic forecasts for reservoir management is, for example, implemented by Maurer and Lettenmaier (2004). The authors study the influence of synthetic seasonal 12-month hydrological forecasts on the management of six reservoirs in the Missouri River basin in North America. Synthetic forecasts are created by applying an error to past observed flows





(flows reconstructed over a 100-year period). The error is defined according to the lead time (increasing error with lead time) and according to the level of predictability the authors wanted to give to the synthetic forecast. Predictability is assessed by

the correlation between past observed seasonal mean flows and the river basin initial conditions. Four levels of predictability are defined according to the variables considered in assessing the initial conditions: (1) good predictability (climate variables, snow water equivalence and soil moisture are considered); (2) average predictability (only climate variables and snow water equivalence are considered); (3) poor predictability (only climate variables are considered), and (4) zero predictability (no variables are considered). These levels of predictability are expressed in terms of coefficients (the stronger the coefficient, the

higher the predictability), which are taken into account in the error of the synthetic forecast. The results of this study draw two main conclusions: (1) synthetic forecasts with better predictability generate the highest revenues (closer to those of a perfect forecast system), and (2) the size of the reservoir influences the value of the synthetic forecast (for a large reservoir, the difference between the benefits of the synthetic forecast with zero predictability and those of the perfect forecast (observed streamflows) system are 1.8%, against 7.1% for a reservoir reduced by nearly a third of its capacity, which represents a

difference of 25.7 million euros in average annual revenues).

Lamontagne and Stedinger (2018) presented two statistical models to generate synthetic forecast values based on a time series of observed weekly flows. The synthetic forecasts are then used in a simplified conceptual reservoir operation framework where operators aim at keeping their reservoir level at a target level during summer. Hydropower benefits are maximized based on the weekly flows and benefits are computed based on the assumption of constant electricity prices. Forecast quality is

evaluated using the coefficient of determination as a measure of skill. The study showed that more accurate forecasts result in higher reservoir freeboard levels and reduced spills. It also highlighted the importance of using synthetic forecasts with varying precision to compare the relative merit of different forecast products.

Arsenault and Côté (2019) investigated the effects of seasonal forecasting biases on hydropower management. The study is based on the Ensemble Streamflow Prediction (ESP) method, which uses historic precipitation and temperature time series to

build possible future climate scenarios and force a hydrological model. The forecasts are issued at 120-day time horizons. The authors apply a correction factor to the ESP hydrometeorological forecasts, generating a positive bias of +7% (overestimation) and a negative bias of -7% (underestimation). The study is carried out on the Saguenay-Lac-St-Jean hydroelectric complex in Quebec, which consists of five reservoirs. The authors also vary the management constraints, by imposing, or removing, a minimum production constraint on management. The study concludes that more constrained systems tend to be more robust to

forecast biases due to their reduced degree of freedom to optimize the release/storage scheduling of inflows. Forecasts with a positive bias (overestimation) led to lower spill volumes than forecasts with a negative bias (underestimation). In addition, it was shown that forecasts with a positive bias (overestimation) were correlated with a lower reservoir level.

Most of the studies in the literature deal with seasonal forecasts, specific flood events or specific contexts of application, including single-site case studies. Furthermore, to the best of our knowledge, there have not been studies that tried to untangle

the influence of the different quality attributes of a forecast on the management of reservoirs. Most of the existing studies either conclude on an overall link between the quality of hydrological forecasts and their economic value, without specifying which quality attribute has the greatest influence on the economic value, or focus on a single particular attribute. For instance, Stakhiva





and Stewart (2010) show that improving the reliability of hydrological forecasting systems can improve hydroelectric reservoir management. Côté and Leconte (2016) also mention the negative impact of under-dispersion of a hydrological forecasting
system on reservoir management. Other studies focus on impacts of forecast accuracy on reservoir management, particularly when dealing with extreme events, such as floods or droughts (Turner et al., 2017; Anghileri et al., 2016; O. Kim et al., 2007).

   The aim of this paper is to present a study that investigates the impact of quality attributes of short-term (7 days ahead) hydrological forecasts on the management of hydroelectric reservoirs under different inflow conditions. For this, we present a method for creating synthetic hydrological forecasts of controlled quality and we apply the different forecasting systems
generated to a reservoir management model. The model, based on a linear optimization algorithm, is designed to represent conceptual reservoirs and management contexts, with a simplified parametrization that takes into account hypothetical reservoir physical parameters and the actual inflow variability from the upstream catchment area. This framework allows us to investigate several sites and untangle the influence of different attributes of forecast quality on hydroelectric production revenues. Our study is based on data from 10 catchments in France. In the following, Section 2 presents the case study areas, the data and
methods used. Section 3 presents the results and discussions and is followed by Section 4, where conclusions are drawn.

## 2    Data and methods

### 2.1    Case study areas and streamflow data

This study is based on a set of 10 catchments located in the Southeast of France. They were selected to represent a variety of hydrological regimes and areas where the French electric utility company EDF operates, or has interest in operating, hydro-
electric dams. Figure 1 shows the location and hydrological regimes of the studied catchments. Catchments 1, 9 and 10 are located in the French Alps. They are described by a snow-dominated hydrological regime, with peak flows observed in spring due to snow melt. Catchment 7 is located in the Jura mountains and its hydrological regime is dominated by peak flows in winter, followed by high flows also in spring. The same is observed for catchments 2, 3, 4, 5, 6 and 8, located in the Cévennes mountains, where the hydrological regime is also marked by very low summer flows.
Daily streamflow data come from the French database "Banque HYDRO" (Leleu et al., 2014). They are either natural flows or, when the flows are influenced by existing dams, naturalized flows at the catchment outlet. In this study, they represent the inflows to the hydroelectric dams and are used to create the synthetic hydrological forecasts of different quality for the period 2005-2008.





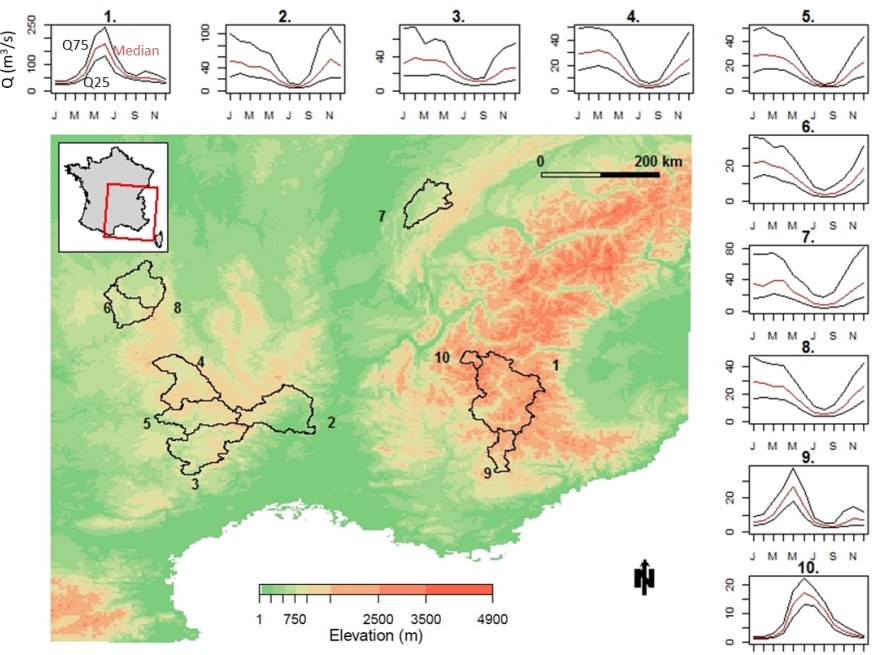

**Figure 1.** Location and hydrological regimes of the 10 studied catchments in France. Lines represent the $75^{th}$ (upper black line), $50^{th}$ (central red line) and $25^{th}$ (lower black line) percentiles of interannual daily flows (in m$^3$/s), evaluated with observed streamflow data available for the period 1958-2008.

## 2.2 Generation of synthetic hydrological forecasts

In order to investigate the impact of the quality of the forecasts on the management of hydroelectric reservoirs, we created time series of 7-day ahead synthetic daily streamflow forecasts of controlled quality for each studied catchment. For this, for a given day and lead time, we first generate a reliable 50-member ensemble forecast based on the observed daily streamflow value and a parametrized log-normal distribution, and then we introduce biases on the generation processes. For a given day and lead time, the approach can be described by the following two major steps:

1. **Creation of synthetic reliable forecasts:** We consider the synthetic ensemble forecast probability distribution as a log-normal distribution with 2 parameters: mean ($\mu$) and standard deviation ($\sigma$). The standard deviation parameter is set as a function of a spread coefficient ($D^2$) and the mean. In other terms, $\sigma$ is expressed by a multiplicative error around the mean (eq. 1). The higher the spread coefficient, the higher the standard deviation.

$$\sigma = D^2 \times |\mu| \qquad (1)$$

The location parameter mean ($\mu$) depends on the daily observed streamflow. In probability theory, if a value $X$ follows a log-normal distribution with parameters ($\mu$, $\sigma$), the variable $Y = log(X)$ follows a normal distribution with parameters ($\mu$, $\sigma$). The variate $Z = \frac{Y-\mu}{\sigma}$ follows a standard normal distribution of parameters $\mu$=0 and $\sigma$=1. For a probability $0 < p$





< 1, the quantile function of the standard normal distribution returns the value $z$ such that:

$$F(z) = Pr(Z \leq z) = p \tag{2}$$

Considering that the variable $X$ represents the observed streamflow, with $Y = log(X)$, the quantile $qp$ associated with the probability $p$ is then:

$$\frac{log(X) - \mu}{\sigma} = qp \tag{3}$$

From (eq. 1), the log-normal mean $\mu$ is then given by:

$$\mu = \frac{log(X)}{1 + qp \times D^2} \tag{4}$$

In order to guarantee the creation of a reliable ensemble forecast, values of $p$ are drawn randomly between 0 and 1 from a uniform probability distribution, for each day and lead time. The sampling must thus be large enough to achieve equal selection of probabilities and obtain a reliable ensemble forecast. Finally, from the log-normal distribution with mean ($\mu$) and standard deviation ($\sigma$), we randomly draw 50 members to generate an ensemble.

The steps above are carried out for each day of forecast and each lead time independently, which means that temporal
correlations can thus be lost. To retrieve correlated 7-day trajectories, we apply an approach based on the Ensemble Copula Coupling (ECC) post-processing methodology (Schefzik et al., 2013; Bremnes, 2008). The approach consists of rearranging the sampled values of the synthetic forecasts in the rank order structure given by a reference ensemble forecasting system, where physically-based temporal patterns are present. In our study, the operational ensemble forecasting system produced at the forecasting centres of EDF is taken as reference. Their operational system is based on a concep-
tual rainfall-runoff model (MORDOR model; Garçon, 1996) forced by the 50 members of the meteorological ensemble forecasting system issued by the European Centre for Medium-Range Weather Forecasts (ECMWF). It produces 7-day ahead streamflow forecasts daily. Forecast members are ranked and the rank structure is applied to the synthetic forecasts to create reordered trajectories that match the temporal evolution of the reference forecasting system.

2. **Introduction of biases:** From the previous step, we can generate ensemble forecasts that are reliable, sharp (very close
to the observed streamflows) and unbiased. To deteriorate the quality of the forecasts, we implement perturbations to be added to the generation process.

To deteriorate the reliability of the synthetic forecasts, the values of $p$, which define the position of the observation in the probability distribution of the ensemble forecast, are not taken randomly. This is done by introducing a reliability coefficient (R) as a power coefficient in the $p$ value: $p^R$. According to the value taken by R, the random drawing will be
biased (R $\neq$ 1) or not (R = 1). We created synthetic ensembles with a negative bias (0 < R < 1) and a positive bias (R > 1), which are associated with forecasts that underestimate and overestimate the observations, respectively.

To generate under-dispersed ensembles, the synthetic generation is controlled so that high flows are underestimated and low flows are overestimated. In practice, each daily observed streamflow is compared with the quantiles 25% and 75%





of the observed probability distribution. If the daily observation is lower than the quantile 25%, the $p$ value is forced to
be between 0 and 0.1. If the daily observation is higher than the quantile 75%, the $p$ value is forced to be between 0.9
and 1. In this way, 50% of the daily synthetic forecasts are forced to under- or over-estimate the observations.

Finally, to deteriorate the sharpness and the accuracy, the spread coefficient (D) is increased. Higher spread coefficients
will generate less sharp and accurate ensemble forecasts. An ensemble forecast very close to the observed streamflows
is generated from the synthetic ensemble forecasting model with a D coefficient equal to 0.01, corresponding to a spread
factor of 0.01%. We then generated additional ensembles with D coefficients equal to 0.1, 0.15 and 0.2, which correspond
to spread factors of 0.01%, 1%, 2.25% and 4%. These are low values, comparatively to actual biases that can be found in
real-world forecasts. However, since our synthetic generation model is based on a log-normal distribution, the degree of
skewness can increase fast as we increase D (and, consequently, as $\sigma$ is increased), generating streamflows that are too
high to be realistic and used in the reservoir management model.

In summary, for each studied catchment and over the 4-year study period, we generated a total of 16 synthetic ensemble
forecasting systems represented on Table 1 (4 main types to characterize biases × 4 spread factors to characterize sharpness).

| | Bias characterization | | | | | | | | | | | | | | | |
|---|---|---|---|---|---|---|---|---|---|---|---|---|---|---|---|---|
| Sharpness characterization | UnB: Reliable | | | | OvE: Overestimation | | | | UnE: Underestimation | | | | UnD: Underdispersed | | | |
| Spread factor (%) | 0.01 | 1 | 2.25 | 4 | 0.01 | 1 | 2.25 | 4 | 0.01 | 1 | 2.25 | 4 | 0.01 | 1 | 2.25 | 4 |

**Table 1.** Summary of the sixteen synthetic forecasting systems generated, according to the four bias characterizations (UnB, OvE, UnE, UnD) and the four sharpness characterizations (spread factors) applied

Each synthetic ensemble forecasting system was generated daily, with 50 members and up to 7 days of forecast lead time.
Figure 2 shows examples of the four synthetic ensemble forecasts generated for the 1-day lead time at the catchment 1 (Fig.1)
for the year 2005. It shows:

- UnB (Reliable and unbiased, Figure 2.a): forecasts uniformly distributed around the observed flows,

- OvE (Overestimation, Figure 2.b): systematic positive biases towards forecasts that overestimate the observed flows,

- UnE (Underestimation, Figure 2.c): systematic negative biases towards forecasts that underestimate the observed flows,

- UnD (Underdispersed, Figure 2.d): flows greater than the historic $75^{th}$ percentile are underestimated, and flows lower
  than the historic $25^{th}$ percentile are overestimated.





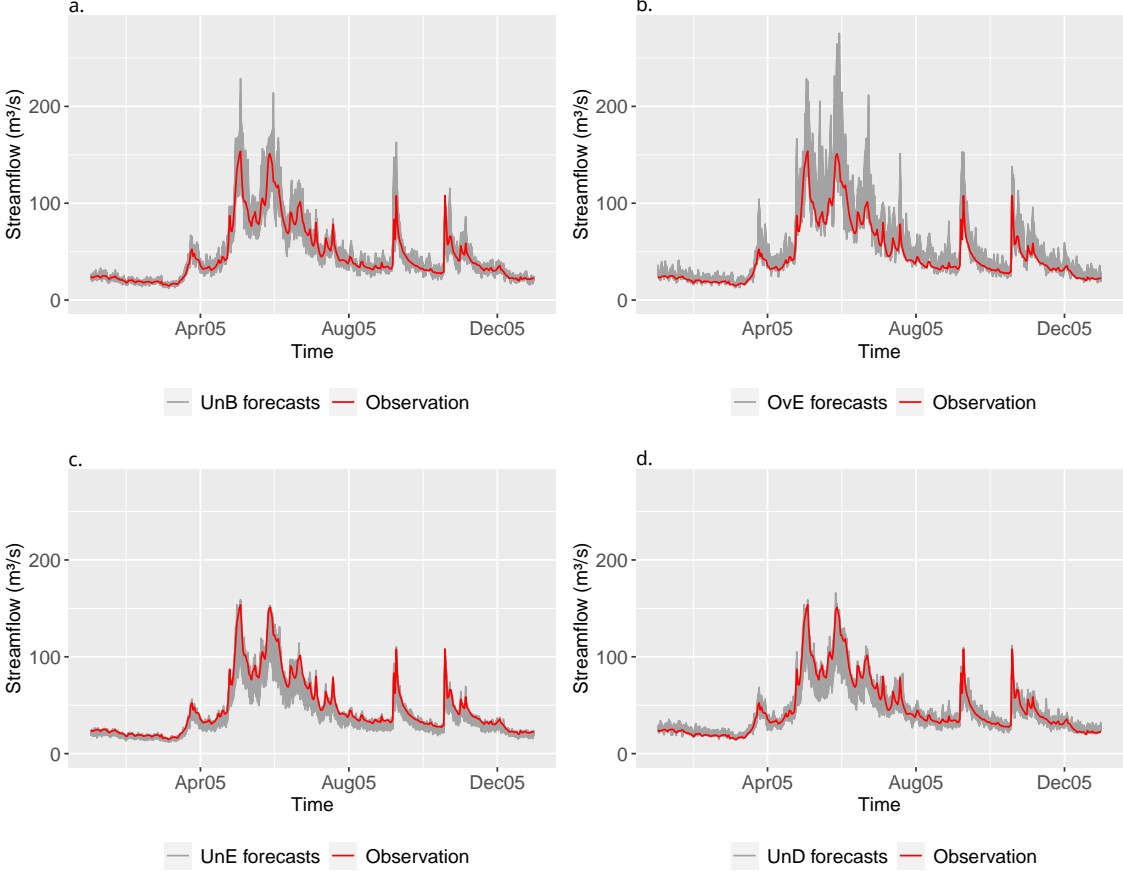

**Figure 2.** Illustration of the synthetic forecasts generated: a. UnB (reliable and unbiased forecasts), b. OvE (forecasts that overestimate), c. UnE (forecasts that underestimate), d. UnD (underdispersed forecasts). The example is for catchment 1 (Fig. 1) and the 1-day forecasts of 2005. The spread factor used for this figure is 2.25%.

### 2.3 Reservoir management model

The reservoir management model is based on linear programming (LP) to solve the optimization problem: maximize hydropower revenue under the constraints of maximum and minimum reservoir capacities. Linear programming is one of the simplest ways to quickly solve a wide range of linear optimization problems. The linear programming model used in this study was developed in collaboration with EDF. It is an improvement of a heuristic model of hydropower reservoir inflow management that was previously designed for research purposes. It defines the reservoir management rules at the hourly time step, based on deterministic 7-day inflow forecasts and hourly time series of electricity prices.

Electricity prices in France have seasonal variability, with higher prices in winter due to the higher demand of electricity for heating. They also vary within a week (lower prices are observed during weekends when industrial demand is lower) and within a day (higher prices are observed during demand peak demand hours). In this study, we use the hourly energy price





time series for the period 2005-2008 from the EPEX-SPOT market (https//www.epexspot.com/fr/). This study period avoids the negative market prices observed after 2008. Since we want to isolate the influence of the quality of inflow forecasts on the revenue, we used observed prices, instead of forecast prices.

The ten studied catchments define the inflows to ten hydroelectric reservoirs, which are conceptually parametrized as following: given the focus of the study on 7-day inflow forecasts, the storage capacity of each reservoir is defined as five times 235 the historic mean daily flow; the maximum electricity production capacity, which is related to production power, is set at three times the historic mean daily flow; the minimum storage capacity is set at 0 Mm$^3$ for each reservoir. Given the historic mean daily flows of the studied catchments, the conceptual sizes of the reservoirs vary between 3.18 Mm$^3$ and 34.22 Mm$^3$ in this study. These dimensions are not the actual dimensions of the reservoirs, although the inflows from the synthetic hydrological forecasts reflect the actual hydrological variability. Since the reservoir management model is a deterministic model, the 240 members of the synthetic ensemble forecasts generated are averaged and only the ensemble mean is used.

The LP optimization model defines, each day, an optimal release sequence (operation scheduling), which amounts to the water to be used to produce electricity. A rolling-horizon optimization scheme is used. At each day, the optimization problem is solved considering a 7-day window. It is informed by the 7-day synthetic streamflow forecast (ensemble mean) and the hourly electricity prices. The algorithm maximizes the hydropower revenue, searching for an optimal release sequence over 245 the week. It tries to use all the incoming volume to produce electricity, maximizing thus the immediate benefits of electricity generation.

The main objective of the reservoir management model is to maximize hydropower revenue while meeting the constraints of the reservoir. The management objective is quantified according to the objective function below, which is maximized over a week for each day of the study period:

$$max \sum_{h=0}^{H-1} p_h \times \rho \times q_h \tag{5}$$

where $h$ refers to the hour of the week (in total, $H = 168$ hours); $p_h$ refers to the hourly electricity price at hour $h$; $\rho$ refers to the efficiency of the power plant, in MWh/m$^3$.s$^{-1}$, which is a constant equal to 1 MWh/m$^3$.s$^{-1}$ in this study; $q_h$ refers to the release in m$^3$/s used for the production at hour $h$.

Mayne et al. (2000) classify management constraints into two categories: hard constraints and soft constraints. Dobson 255 et al. (2019) define these constraints as follows: *"Hard constraints are those constraints that cannot be violated under any circumstance and typically represent physical limits [...] Soft constraints, instead, are those constraints that should not be violated but that are not physically impossible to break."*. In our experiment, the maximum capacity is a soft constraint, while the minimum capacity is defined as a hard constraint. When a major event occurs and the reservoir does not have the storage capacity required to store the inflow volume, the maximum reservoir capacity constraint is violated. The storage constraints 260 and the temporal evolution of the stock are quantified as:

$$v_h^{min} \leq v_h \leq v_h^{max} \tag{6}$$





$$v_h = v_{h-1} + K(a_h - q_h) \tag{7}$$

where $v_h^{min}$ and $v_h^{max}$ represent, respectively, the minimum and maximum volume of the reservoir at hour $h$ in Mm$^3$; $a_h$ represents the forecast inflow at hour $h$ in m$^3$/s; $K$ represents the conversion constant from m$^3$/s to Mm$^3$/h (equal to 0.0036).

The optimization is also constrained by the maximum production capacity, which is considered as a hard constraint. Production therefore cannot exceed the maximum production capacity:

$$0 \leq q_h \leq q_{max} \tag{8}$$

where $q_h$ refers to the release in m$^3$/s at hour $h$; $q_{max}$ refers to the maximum release (associated to the maximum production capacity).

Furthermore, the optimization is constrained by the weekly release for electricity production, which cannot be higher than the weekly inflows. This constraint is considered as a soft constraint. It has been implemented for this research management model and is not representative of the real operational constraints. It can be expressed as:

$$\sum_{h=0}^{H-1} (K \times q_h) \leq A = \sum_{h=0}^{H-1} (K \times a_h) \tag{9}$$

where $A$ represents the cumulative weekly inflows.

When applying the model, after the optimization phase, once the operation schedule is defined for the coming week, the model simulates the management of the reservoir with the actual observed inflows over the release schedule defined for the first 24 hours. This simulation phase consists in applying the optimal command of releases obtained during the optimization phase to the first day (there is no re-optimization). At this phase, it may happen that the observed inflows are very different from the forecast inflows used for the optimization, and, due to the management constraints, it is not possible to follow the

optimal command. In this case, the rule is modified to allow the operations to be carried out within the storage constraints. When the management rule at one hour $h$ induces a non-respect of the minimum storage constraint, the release is decreased until the constraint is respected. On the other hand, when a volume of water is spilled, the release is increased. We note that, in this case, the modified management rule does not always avoid discharge and a volume of spilled water may occur.

The volume obtained at the end of the 24-hour simulation phase is used to update the initial volume of the reservoir for

the next forecast day and optimization. This is done in a continuous loop over the entire 4-year study period (2005-2008), for each catchment and for the 16 synthetic streamflow forecasts of different forecast quality. At the end, the amount of hourly electricity produced is multiplied by the price (in Euro/MWh) to obtain the revenue. The impact of forecast quality on the revenue (economic value) is then assessed.

## 2.4   Evaluation of forecast quality

In terms of forecast quality, we focus on assessing the reliability, sharpness, bias and accuracy of the forecasts.





The reliability is a forecast attribute that measures the correspondence between observed frequencies and forecast probabilities. It can be measured by the probability integral transform (PIT) diagram (Gneiting et al., 2007; Laio and Tamea, 2006), at each forecast lead time. The diagram represents the cumulative frequency of the values of the predictive (forecast) distribution function at the observations. A reliable forecast has a PIT diagram superposed with the diagonal (0-0 to 1-1). It means that

the observations uniformly fall within the forecast distribution. A forecasting system that overestimates the observations is represented by a curve above the diagonal. If the PIT diagram is under the diagonal, it indicates that observations are systematically underestimated. A PIT diagram that tends to be horizontal means that the forecasts suffer from under-dispersion (i.e. observations often fall in the tail ends of the forecast distribution). At the opposite, a PIT diagram that tends to be vertical means that the forecasts are over-dispersed.

The sharpness of a forecast corresponds to the spread of the ensemble forecast members. It is an attribute independent of the observations, which is therefore specific to each forecasting system. To evaluate the sharpness of a forecast, for each lead time, the 90% inter quantile range (IQR) can be used (Gneiting et al., 2007). It corresponds to the difference between the quantile $95^{th}$ and the quantile $5^{th}$ of all ensemble members. The IQR score is evaluated for each forecast day and then averaged over the entire study period. The smaller the IQR, the sharper the forecast.

The forecast bias measures the average error of a forecast in relation to the observation over a given time period. Bias measurement is used to detect positive (forecasts greater than observations) or negative (forecasts lower than observations) biases. To evaluate the bias of the synthetic ensemble streamflow forecasts, the percent bias (Pbias) is computed for each day $i$ and lead time (Fan et al., 2016; Waseem et al., 2015). It compares the daily observation $o$ with the daily mean of the ensemble forecast $m$. The Pbias score is then averaged over the entire study period of $N$ forecast days:

$$Pbias = 100 * [\frac{\sum_{i=1}^{N}(m_i - o_i)}{\sum_{i=1}^{N} o_i}] \qquad (10)$$

The Pbias is negative (positive) when forecasts underestimate (overestimate) the observations.

The accuracy of a forecast represents the difference between an observed value and an expected forecast value. It is often assessed with the root mean square error (RMSE), which corresponds to the square root of the mean square error. In the case of an ensemble forecast, the average of the ensemble forecast is often used to assess the accuracy of the forecasting system. In

this study, the RMSE is normalized with the standard deviation (SD) of the observations to allow comparison among different catchments:

$$NRMSE = \frac{RMSE}{SD_{obs}} \qquad (11)$$

The lower the NRMSE, the more accurate the forecast is.

Finally, we evaluated the overall forecast quality of each forecasting system with the continuous ranked probability score

(CRPS) (Hersbach, 2000). It compares the forecast distribution to the observation distribution (a Heaviside step function at the observation location) over the evaluation period. The score is better if the probability distribution of the forecasts is close to that of the observations. The lower the CRPS, the better the forecasts are. In this study, the CRPS is also normalized with the





standard deviation (SD) of the observations to reduce the impact of the catchment size on this score (Trinh et al., 2013):

$$NCRPS = \frac{CRPS}{SD_{obs}} \qquad (12)$$

## 325  3  Results and discussions

### 3.1  Quality of the generated synthetic hydrological forecasts

In order to validate the model used to generate synthetic forecasts of controlled quality, we evaluated the quality of each ensemble hydrological forecasting system in terms of sharpness (Fig. 3), reliability (Fig. 4), systematic bias (Fig. 5), accuracy (Fig. 6) and overall quality (Fig. 7). The quality of the synthetic ensemble forecasts is presented at the 1-day lead time only, since, by construction, the quality of the synthetic forecasts does not vary according to the lead time. In all the figures, except Fig. 4, the top graphs show, in the form of boxplots (maximum value, percentiles 75, 50 and 25, and minimum value), the distribution of the scores of quality for the 10 catchments of the study. The bottom graphs highlight the evolution of the median score (percentile 50). In both graphs, the results are presented for four spread factors (0.01%, 1%, 2.25% and 4%).



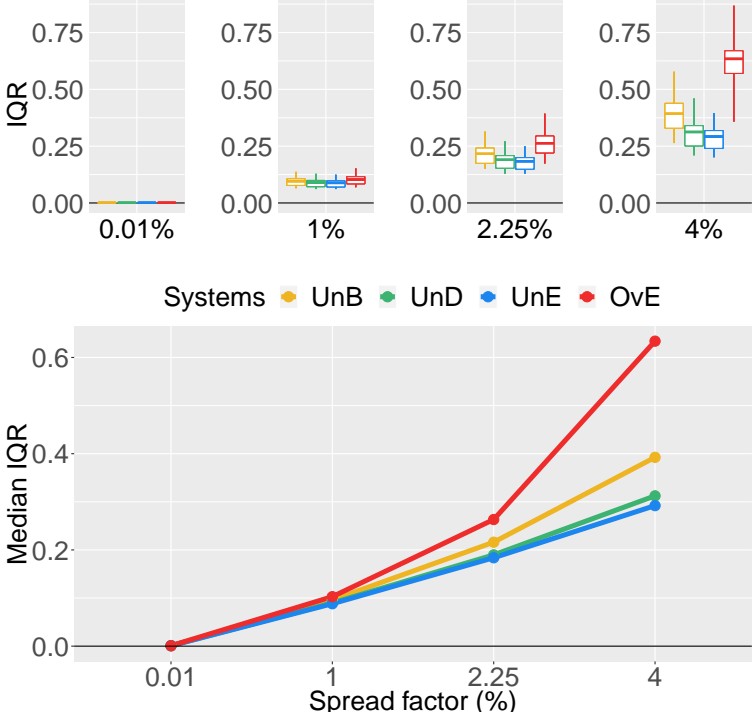

**Figure 3.** IQR score for 1-day ahead synthetically generated forecasts of different quality: unbiased system (UnB, yellow), under-dispersed system (UnD, green), under-estimating system (UnE, blue) and over-estimating system (OvE, red). Systems are based on ensembles generated with four different spread factors (0.01%, 1%, 2.25%, 4%). Top: boxplots represent the maximum value, percentiles 75, 50 and 25, and minimum value over 10 studied catchments. Bottom: median score value.

Figure 3 shows the effect of the increase in the spread factor on the sharpness score (IQR) of each synthetic ensemble
forecasting system. The evolution of IQR values for the unbiased system (UnB) can be used as a reference of expected impacts: the dispersion of the ensemble system increases when the spread factor increases. Therefore, the model designed for the generation of synthetic ensemble forecasts of controlled quality works as expected in terms of spread variations. Additionally, we also observe that the IQR scores are not very different among the forecasting systems and the studied catchments for the smaller spread factors. To differentiate the ensemble systems in terms of their sharpness, it is thus necessary to have a spread
factor greater than 1% in the synthetic forecast generator model, regardless of the catchment location or the bias of the system. The inter-catchment difference as well as the differences among the forecasting systems increase as the spread factor increases. These differences also reflect the way the synthetic ensemble forecasts were generated. The implementation of a reliability bias towards overestimation of streamflows (OvE system) has a strong impact on sharpness. IQR score values are the highest for this system, particularly when the spread factor is high. The spread of the forecasts is thus the highest for this system. This can
be explained by the fact that, by construction, there is no physical upper limit imposed to the overestimation of streamflows. On the contrary, for the system that was generated to present a bias towards undestimation (UnE), the lower limit physically





exists and corresponds to zero flows. This explains why this system shows low IQR scores, even at the highest spread factor. In the under-dispersed system (UnD), the low flows are overestimated, while the high flows are underestimated. By construction, the forecasts are thus more concentrated and the dispersion of this system tends to be small. This is also reflected in Fig. 3,
where the values of IQR for the UnD system are very close to those of the UnE system.

The evaluation of reliability is shown in Fig. 4. The PIT diagram of each catchment is represented (lines) for the higher spread factor only (4%), when the differences in the quality of the forecasting systems are higher. The lines in the PIT diagram clearly show the effectiveness of the forecast generator model to introduce reliability biases in the unbiased forecasting systems of all catchments. The cumulative distributions of the PIT values of the unbiased systems (UnB) are located around the diagonal,
showing a uniform distribution of the PIT values, as expected in a probabilistically calibrated ensemble forecasting system. The forecast deficiencies of the biased-generated systems are illustrated in the PIT diagram by their distance to the uniformity of a reliable system. The forms of the curves reflect well the under-dispersion of the system UnD, as well as the overestimation and the underestimation of the systems OvE and UnE, respectively.

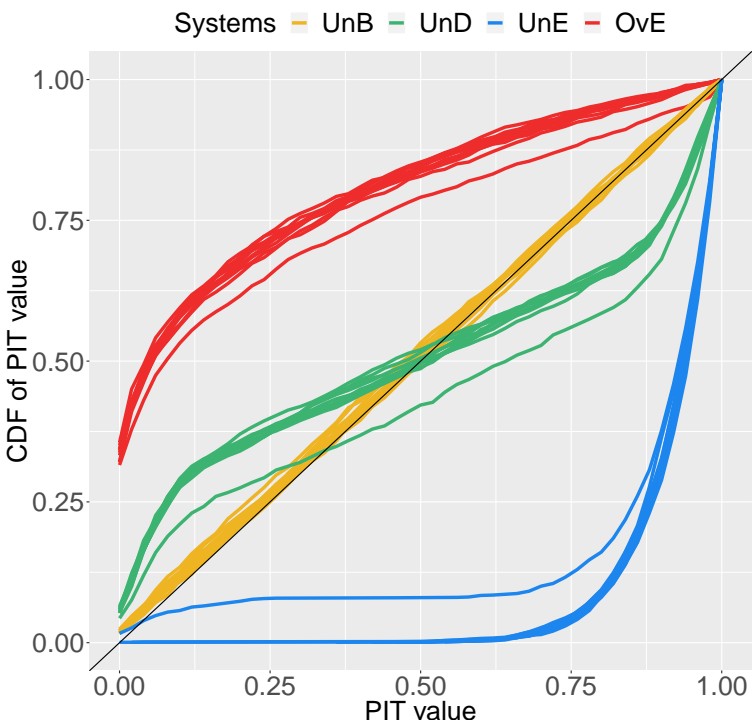

**Figure 4.** PIT diagram for 1-day ahead synthetically generated forecasts of different quality: unbiased system (UnB, yellow), under-dispersed system (UnD, green), under-estimating system (UnE, blue) and over-estimating system (OvE, red). For each system, each line represents one of the 10 studied catchments. Forecast are based on ensembles generated with a 4% spread factor.





The reliable system (UnB) also shows zero to very low percent bias, as illustrated in Fig. 5. For this system, a slight positive

bias appears when using a 4% spread factor. The fact that the increase in spread leads to a slight positive bias (overestimation), even when the system is generated to be probablistically unbiased, may be the consequence of the skewness of the log-normal distribution used in the forecast generation model, which increases as the spread increases and may result in the generation of some very high values, affecting the median bias. This impact is however much smaller comparatively to the impact of adding biases to the reliable forecasts. From Fig. 5, we can see a strong positive bias for the system that tends to overestimate

streamflow observations (37% of median Pbias value for OvE and spread factor of 4%), and a negative bias for the system that tends to underestimate them (up to -18% of median Pbias value for the UnE system). We also observe that there are larger differences in Pbias values among catchments in the OvE forecasting system, particularly at spread factor of 4% (Pbias values vary from 25% to 75%). A negative bias is observed for under-dispersed forecasts (-11.5% of median Pbias value for UnD and spread factor of 4%). This indicates that, in the UnD forecasting system, the impact of the underestimation of high flows in the

percent bias is higher than the impact of the overestimation of low flows. Finally, in all biased generated systems, the higher the spread factor, the higher the absolute value of the Pbias of the system.

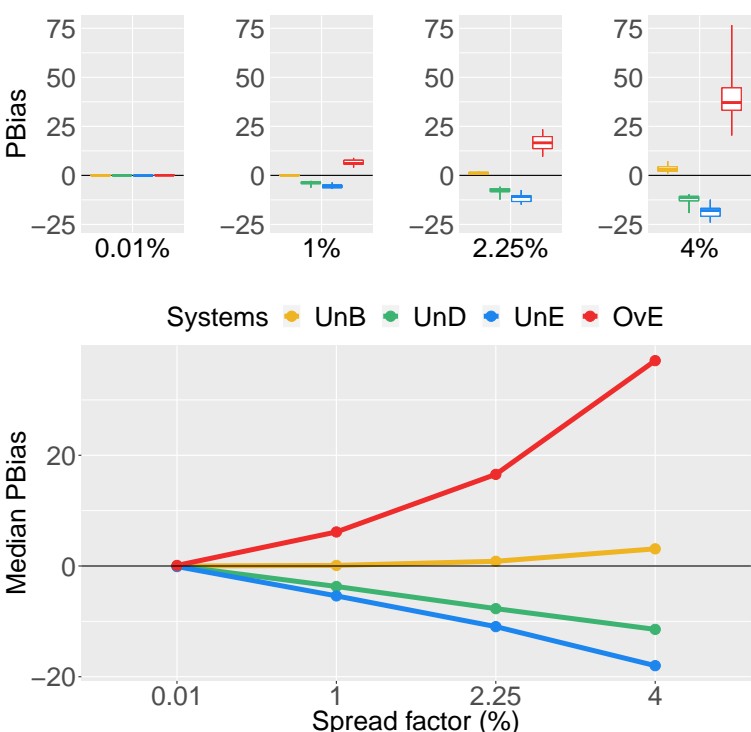

**Figure 5.** Pbias score for 1-day ahead synthetically generated forecasts of different quality: unbiased system (UnB, yellow), under-dispersed system (UnD, green), under-estimating system (UnE, blue) and over-estimating system (OvE, red). Systems are based on ensembles generated with four different spread factors (0.01%, 1%, 2.25%, 4%). Top: boxplots represent the maximum value, percentiles 75, 50 and 25, and minimum value over 10 studied catchments. Bottom: median score value.





The evaluation of the scores NRMSE (accuracy, Fig. 6) and NCRPS (overall forecast quality, Fig. 7) also illustrates the impact of introducing biases in a reliable ensemble streamflow forecasting system. Overestimation leads to the worst scores and the highest differences in performance among catchments. Systems generated with under-estimation and under-dispersion

biases have very similar scores. For all systems, scores get worse when increasing the spread factor (i.e. the ensemble spread). Although better for the unbiased and reliable system (UnB), the accuracy score (NRMSE) does not strongly differentiate the UnB, UnD and UnE systems in terms of accuracy of the ensemble mean. This may be linked to the fact that the RMSE is based on absolute values of the errors, and is not sensitive to the direction of the error (as in Pbias). Also, the fact that larger differences have a larger effect on RMSE, given that it is based on the square root of the average of squared errors, the high

streamflow values generated in the unbiased system when using the higher spread factor penalize this system, leading to RMSE scores very close to the scores of the biased UnD and UnE systems, where high streamflow forecast values tend to occur less often.

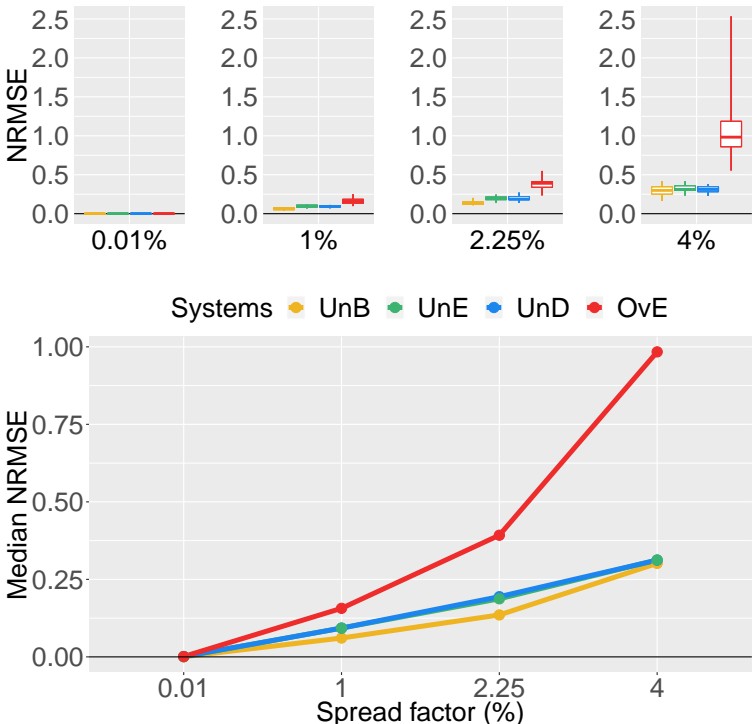

**Figure 6.** NRMSE score for 1-day ahead synthetically generated forecasts of different quality: unbiased system (UnB, yellow), under-dispersed system (UnD, green), under-estimating system (UnE, blue) and over-estimating system (OvE, red). Systems are based on ensembles generated with four different spread factors (0.01%, 1%, 2.25%, 4%). Top: boxplots represent the maximum value, percentiles 75, 50 and 25, and minimum value over 10 studied catchments. Bottom: median score value.





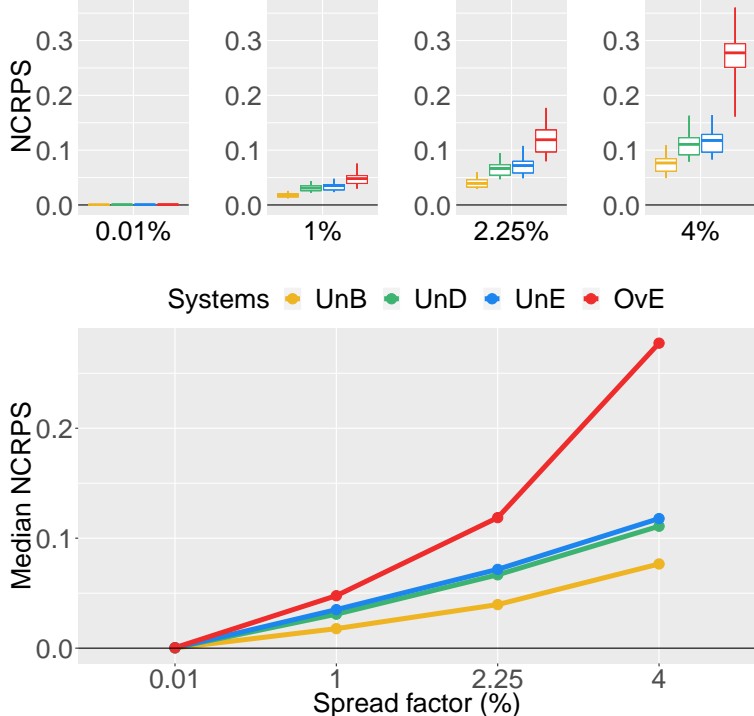

**Figure 7.** NCRPS score for 1-day ahead synthetically generated forecasts of different quality: unbiased system (UnB, yellow), under-dispersed system (UnD, green), under-estimating system (UnE, blue) and over-estimating system (OvE, red). Systems are based on ensembles generated with four different spread factors (0.01%, 1%, 2.25%, 4%). Top: boxplots represent the maximum value, percentiles 75, 50 and 25, and minimum value over 10 studied catchments. Bottom: median score value.

## 3.2 Economic value of the generated synthetic hydrological forecasts

The value of the different forecasting systems is assessed based on the total economic revenue obtained from the hydropower

reservoir operation, when using, on a daily basis, each system's 7-day forecasts as input to the reservoir management model (LP optimization) over the study period (2005-2008). The revenue obtained with each synthetic forecasting system is then evaluated against the revenue obtained using a reference system. This reference system is given by the observed streamflows. It is thus equivalent to a 'perfect forecasting system', where the forecasts are always identical to the observed inflows. Hence, the maximum revenue is obtained by the reference system. The gain in revenue obtained for each synthetic system is expressed as

the percentage gain in relation to the revenue of the reference system (N.Gain in %). This percentage gain is therefore negative (i.e. a percentage of loss in relation to the reference). The results obtained are shown in Fig. 8. The graph on top shows, in boxplots, the distribution of the percentage gains (maximum value, percentiles 75, 50 and 25, and minimum value) of the 10 studied catchments. The graph at the bottom highlights the median gain. Both graphs show the gain for each synthetic forecast system of controlled quality and as a function of the spread factor used to generate the forecasts.



The economic revenues of the different forecasting systems are very similar for the smaller spread factors. The difference in
economic value between the synthetic forecast systems widens with the increase in the spread factor. Moreover, the percentage
gains show a clear tendency to decrease as the spread factor increases. A given forecasting system will lose more revenue in
comparison with the reference 'perfect system' as it becomes more dispersed. This observation is in line with the analysis of
the quality of the systems: as the spread increases, the quality (sharpness, accuracy, reliability and overall quality) decreases
and also the value (percentage gain in revenue) decreases.

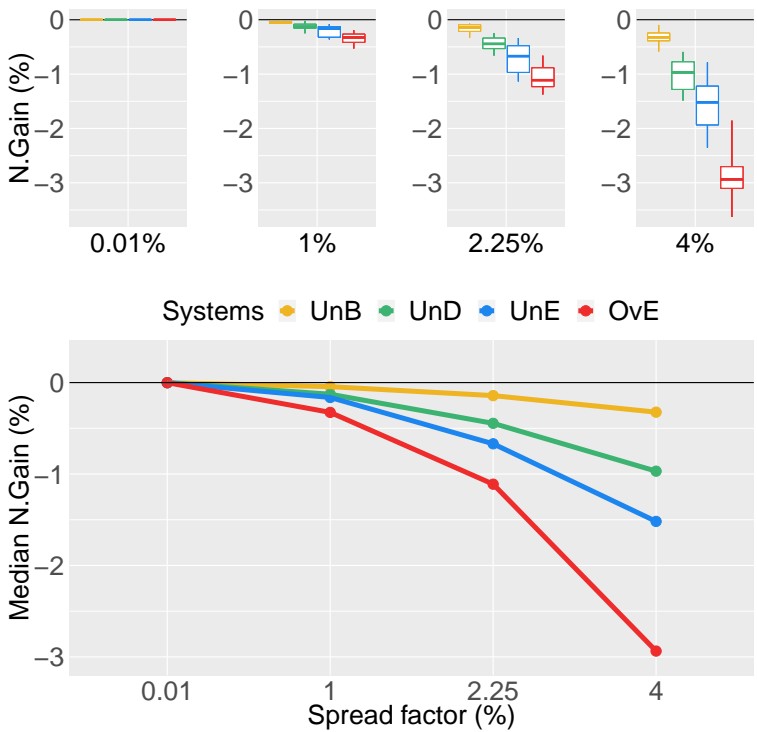

**Figure 8.** Percentage gain in hydropower revenue (N.Gain in %) for synthetically generated forecasts of different quality: unbiased system
(UnB, yellow), under-dispersed system (UnD, green), under-estimating system (UnE, blue) and over-estimating system (OvE, red). Systems
are based on ensembles generated with four different spread factors (0.01%, 1%, 2.25%, 4%). Top: boxplots represent the maximum value,
percentiles 75, 50 and 25, and minimum value over 10 studied catchments. Bottom: median value.

Fig. 8 and the analysis of synthetic forecasts quality (Fig. 3, Fig. 4, Fig. 5, Fig. 6 and Fig. 7) clearly show that the forecasting
system displaying the worst scores in terms of forecast quality is also the one that displays the lowest economic value (system
OvE, red line in Fig. 8). In the ranges and within the conditions of our experiment, a forecast system that, in median values,
overestimates streamflows at about 30% (Pbias) generates a loss in revenue of up to 3%, comparatively to the revenue generated
by a forecasting system that perfectly forecasts the observed inflows to the reservoir. Similarly, the unbiased system (UnB),
which has the best quality among the synthetic forecasting systems (reliability, bias and overall quality), is the system that





provided hydropower revenues the closest to the revenues of a 'perfect system'. The second best system in terms of economic value is the system that suffers from under-dispersion (UnD). Although, in terms of forecast quality, this system ranks closely to the forecast system that underestimates inflows (UnE), it performs, in median values, about 0.5% points better in terms of

economic gains. The systems that under- and over-estimate inflows (UnE and OvE, respectively) are those that show a steeper rate of loss in economic revenue as the spread of the forecasts increases. When moving from a spread factor of 2.25% to a spread factor of 4%, the median percentage gains of the system UnE move from -0.67% to -1.5%, while for the system OvE, it moves from -1% to -3%. Overall, the rank in economic value of the synthetic forecasting systems is similar to their rank in quality, notably according to the Pbias and the NCRPS scores.

**3.3 Influence of forecast bias on the total amount and hours of electricity production**

The economic value of the synthetic forecasts is assessed by the gains of revenue generated when using a given forecasting system as inflow to the reservoir management model. Revenues (in Euros) are calculated by multiplying the hourly electricity production (MW) by the electricity price (Euro/MWh) at the time of production. It is not enough to produce a large amount of electricity to increase revenues. It is also necessary to optimally place the production at the best hours (i.e. when the prices

are higher). Here, we investigate how each synthetic forecasting system influences the total production and number of hours of electricity produced over the study period. Figure 9 shows the normalized total production of each synthetic forecasting system, while Fig. 10 shows the normalized number of hours of production over the entire period. Both are expressed in terms of percentage of the total production (N.Production in % in Fig. 9) or of the total hours of production (N.Hour of production in % in Fig. 10) of the reference system (i.e. the 'perfect system', where forecasts are equal to observations).





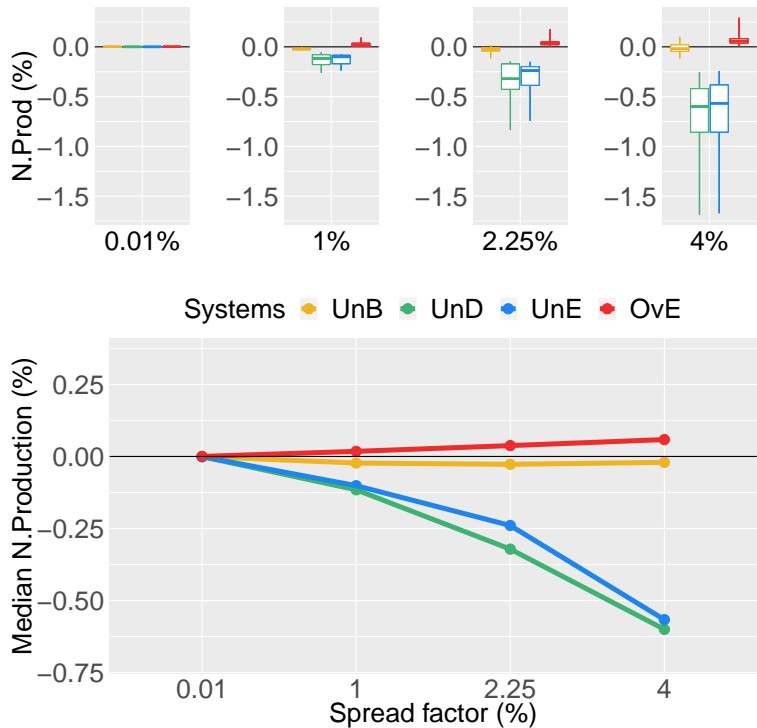

**Figure 9.** Total production of the synthetic ensemble forecasting systems as a percentage of the total production of the reference system (i.e. the 'perfect system', where forecasts are equal to observations): unbiased system (UnB, yellow), under-dispersed system (UnD, green), under-estimating system (UnE, blue) and over-estimating system (OvE, red). Systems are based on ensembles generated with four different spread factors (0.01%, 1%, 2.25%, 4%). Top: boxplots represent the maximum value, percentiles 75, 50 and 25, and minimum value over 10 studied catchments. Bottom: median value.

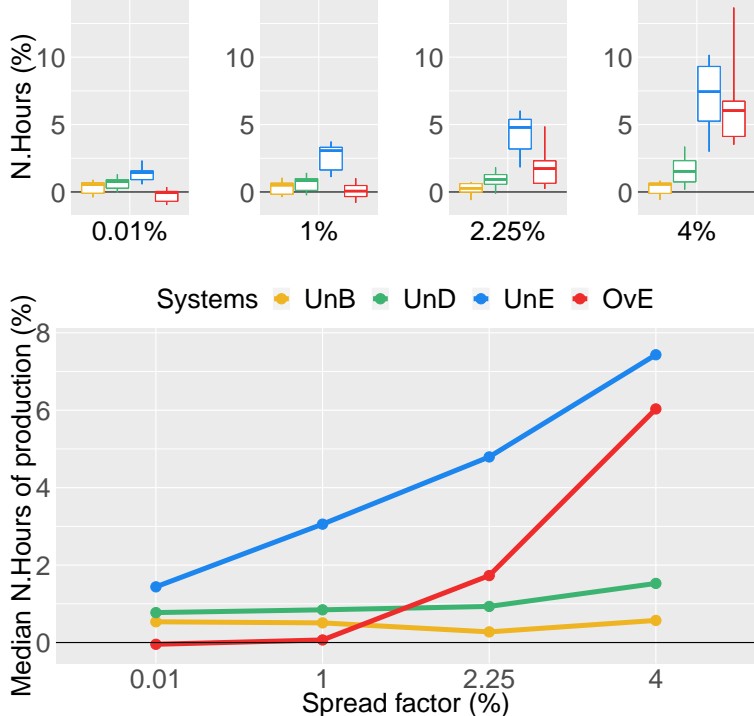

**Figure 10.** Total hours of production of the synthetic ensemble forecasting systems as a percentage of the total hours of production of the reference system (i.e. the 'perfect system', where forecasts are equal to observations): unbiased system (UnB, yellow), under-dispersed system (UnD, green), under-estimating system (UnE, blue) and over-estimating system (OvE, red). Systems are based on ensembles generated with four different spread factors (0.01%, 1%, 2.25%, 4%). Top: boxplots represent the maximum value, percentiles 75, 50 and 25, and minimum value over 10 studied catchments. Bottom: median value.

Comparatively to the reference system, the unbiased forecasting system (UnB) results in lower total electricity production, although differences are very small (Fig. 9). The released volumes (used to produce electricity) are thus very similar to the released volumes of the reference system. In terms of the number of hours of production (Fig. 10), we can see that there are more production hours in the unbiased system than in the reference system (about 0.5% more). This indicates that the unbiased system produces almost the same quantity of electricity, but distributes it over more hours within the study period, which may

reflect a less optimized placement of production than in the reference system. This may explain the tendency of this synthetic forecasting system to show small losses in terms of economic value (Fig. 8), comparatively to the reference system, despite its overall good quality. The increase in the spread factor does not seem to substantially affect the results.

The total production seems to be more sensitive to forecast under-estimation of high flows, as present in the synthetic forecasting systems with underestimation (UnE) and under-dispersion (UnD) biases. These systems show a loss of production

which is sensitive to the spread factor and can reach up to 0.6% in median values, over the 10 studied catchments (Fig. 9). This may be related to the fact that these systems tend to forecast less water flowing into the reservoir. The management model will





therefore plan less releases, which, consequently, tends to lower down the electricity production. The total production losses of these two systems are in opposition to the increase in the number of production hours (Fig. 10). Particularly in the case of the UnE system, production is distributed over up to 7.4% more hours than that of the reference system. It seems that, since

the system forecasts lower inflows than what is observed, the management model has to activate production hours that were not necessarily planned based on the forecasts to release the exceeding inflow that is observed. The optimized production rule has thus to be adapted to the fact that more inflow is observed than forecast. It is done by increasing the number of production hours. We also note that, in percent points, the amount of loss in production is lower than the amount of loss in revenue (gain) for these systems (up to 1% for UnD and 1.5% for UnE, as shown in Fig. 8). The loss in revenue may therefore be explained

by other factors than the loss in total production only, and this loss is not compensated by the increase in production hours. It seems that the way the production is distributed within the hours also plays a role (i.e. how the optimization works in terms of finding the best hours to produce electricity and, consequently, increase the total revenue).

The synthetic forecast system with a positive bias (OvE) is the only one that displays a total production higher than the reference system, although the difference is very small: for a spread factor of 4%, the percentage of increased production is of

only 0.06% (Fig. 9). The OvE system distributes its production over a larger number of hours, but only when the spread factor increases (Fig. 10). The increase of spread also increases the value of the ensemble mean of the OvE system, and, consequently, higher inflows are forecast to the reservoir than later observed. This seems to result in management rules with a higher number of hours of production (i.e. production is activated to release water and lower the levels of the reservoir). However, as in the case of the UnE system, the high number of hours of production does not result in gains in revenue, since the OvE system is

the one with worst economic performance, followed by the UnE system (Fig. 8).

### 3.4 Influence of forecast bias on the rate of electricity production

More production hours does not necessary lead to more total production or more gains in revenue, as seen in the UnE and OvE biased forecasting systems. Here, we investigate how the production is performed within the production hours for these synthetic forecasting systems. In the reservoir management model, production is not always operating at maximum capacity.

For instance, if the maximum production capacity is 200 MW and, at a given hour, only 20 MW is produced, the production rate at this hour is of only 10%. A forecasting system can thus be associated with a large number of production hours, although electricity is finally produced at low rates. Additionally, the gains in revenue are influenced by the electricity prices at the hours of higher production rates. In order to investigate this issue, we considered the UnE and OvE systems with the higher spread factor (4%). We defined four classes of production rates:

– class 1 (C1): production rate less than a quarter of the maximum production capacity,

    – class 2 (C2): production rate between a quarter and a half of the maximum production capacity,

    – class 3 (C3): production rate between a half and three quarters of the maximum production capacity,

    – class 4 (C4): production rate exceeding three quarters of the maximum production capacity.





For each class, we evaluated the number of production hours falling in each class in Fig. 11, and the median electricity
price in Fig. 12 (i.e. the median value of all the prices of all the hours that fall into the given class of production rate). Both
are expressed as differences between the values evaluated when considering the reference system (i.e. the 'perfect system',
where forecasts are equal to observations) and the values obtained with the synthetic forecasting system. Therefore, negative
values indicate that the production hours or median price values of the synthetic forecasting system are higher than those of
the reference system. At the opposite, positive values indicate that the production hours or median price values of the synthetic
forecasting system are lower than those of the reference system. In Fig. 11, the difference in hours within each class is shown,
and in Fig. 12 the differences in median prices within each class is shown. Boxplots represent the maximum value, percentiles
75, 50 and 25, and minimum value over the 10 studied catchments.

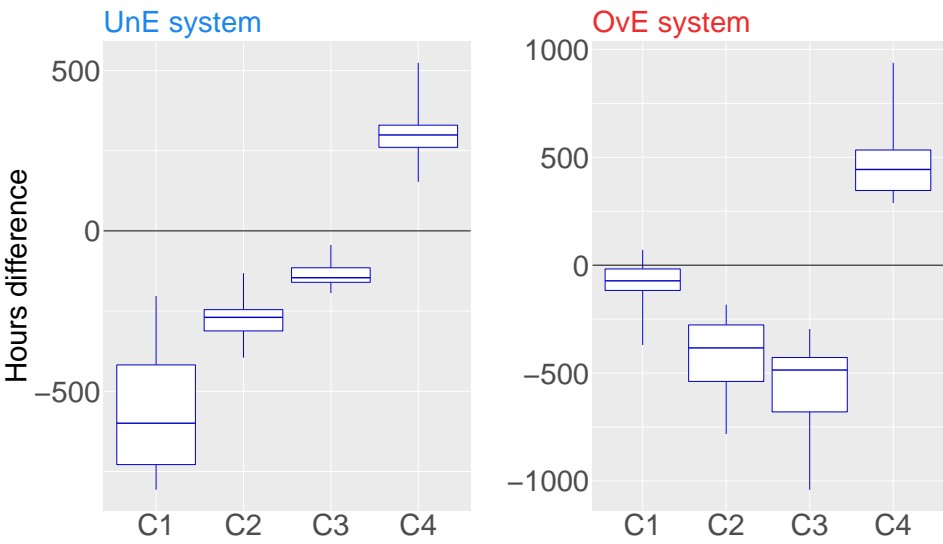

**Figure 11.** Differences in production hours between the reference system (i.e. the 'perfect system', where forecasts are equal to observations)
and two synthetic ensemble forecasting systems: under-estimating system (UnE, blue) and over-estimating system (OvE, red). Differences
are pooled into four classes of production rate (from lower C1 to higher C4 production rate). Synthetic systems are based on ensembles
generated with spread factors 4%. Boxplots represent the maximum value, percentiles 75, 50 and 25, and minimum value over 10 studied
catchments.





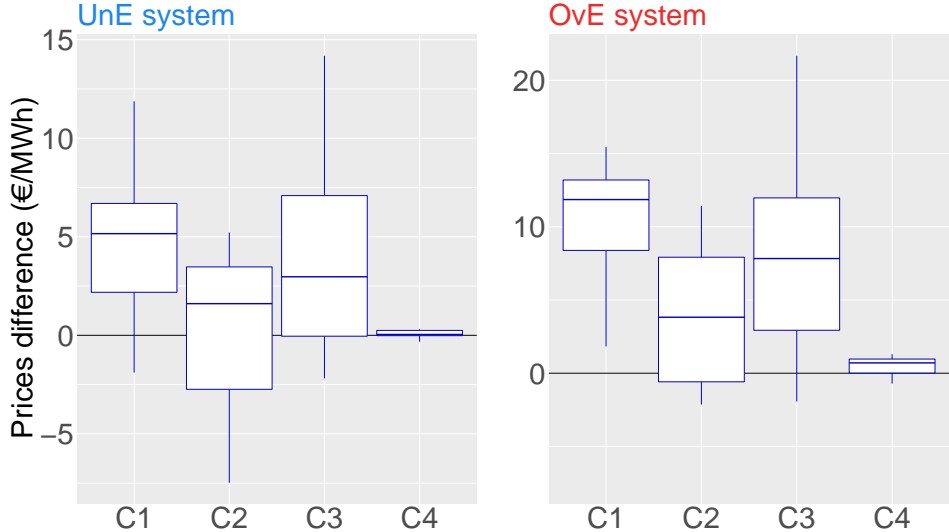

**Figure 12.** Differences in median electricity prices between the reference system (i.e. the 'perfect system', where forecasts are equal to observations) and two synthetic ensemble forecasting systems: under-estimating system (UnE, blue) and over-estimating system (OvE, red). Differences are pooled into four classes of production rate (from lower C1 to higher C4 production rate). Synthetic systems are based on ensembles generated with spread factors 4%. Boxplots represent the maximum value, percentiles 75, 50 and 25, and minimum value over 10 studied catchments.

From Fig. 10, we had seen that the total number of production hours of the OvE and UnE forecasting systems was greater than the total number of hours of the reference system. Fig. 11 shows that production is more frequent in classes of lower

production rate for these systems: both UnE and OvE systems, show more hours than the reference system in classes C1, C2 and C3 (boxplots at negative values), and less hours than the reference system in the most productive class C4 (boxplots at positive values). In median values, system UnE displays 600 hours more than the reference system in the lower class C1, i.e. when production is performed at less than a quarter of the maximum capacity, the UnE system produces more than the reference system. For the OvE system, its performance is notably worse than the reference system for classes C2 and C3 (it

has more hours of production at the C2 and C3 lower rates than the reference system) and for class C4 (in median values, it has about 500 hours less of production at the higher rates than the reference system). The impact of the biased forecasting systems on the performance of the management model is clearly demonstrated. On the one hand, biased forecasts tend to produce more than the reference system when the production rate is low. The reference system counts very few hours of production rate in classes C1, C2 and C3 of low production rate. On the other hand, the biased systems tend to produce less when the production

rate is closer to the maximum production capacity. The difference between the total production of the UnE system and that of the reference system (Fig. 9) is therefore explained by the high frequency of production of the UnE system in class C1 of low production rate, together with their low frequency of production in class C4 of the highest production rate.





Comparatively to the management revenue of the reference system, the losses of the OvE system are the strongest (Fig. 8). This is due to the lower frequency of production at the relatively higher prices of class C4, but also to the fact that, when the

biased system shows more hours of production than the reference system these are at classes of low production rate (notably C2 and C4; see Fig. 11), for which the median prices of electricity are lower than the reference system (Fig. 12). It is therefore possible that the lower economic gains associated with the OvE system are related to the lower average prices of electricity per hour of production. In Fig. 12, we can see that the electricity prices, in median values, used for the assessment of the management revenues of each system, are lower for the UnE and OvE systems than for the reference system, particularly for

the production rate classes C1, C2 and C3 (boxplots at positive values in Fig. 12). Price differences can reach 21.7 €/MWh for the OvE system and the class C3, and 14.2 €/MWh for the UnE system and the class C3. Price differences are almost zero for class C4, when the production rate is at or closer to the maximum capacity. However, at this class, both biased systems, UnE and OvE, have less hours of production. These differences in prices and production hours at high capacity between the biased systems and the reference system may explain their lower economic performance in terms of total management revenue (Fig.

505 8).

### 3.5 Influence of forecast bias on the evolution of stock and on spillage

The reservoir management model evaluates the management revenues based on the electricity produced. The total revenue obtained from using a given forecasting system as inflow forecasts to the reservoir does not take into account how the stock in the reservoir evolves in time, comparatively to the management based on the reference system, where forecasts are equal to

observations. The reservoir management model does not penalize spillage losses. Spillage can be caused by rare and extreme hydrometeorological events, but also by a non-efficient management of the stored water and the reservoir releases. It should be remembered that the reservoirs are conceptual in this study, and have been reduced in size, from their actual sizes, according to their mean annual hydrological inflows. Here, we focus on investigating how the biases in the synthetic forecasting systems impact the evolution of the stock and the spillage losses.

Figure 13 shows the differences in stock between the operation with the reference system and the operation with each of the four synthetic forecasting systems (UnB, OvE, UnE and UnD), with each of the spread factors (0.01%, 1%, 2.25%, 4%). To build this figure, we considered the level of the reservoir at the first hour of each day of the study period. In order to be able to pool the results of the 10 catchments, the differences are divided by the maximum storage capacity of the reservoir that is associated with the catchment. A positive (negative) value of the difference (N.stock error in Fig. 13) indicates that the stock

obtained with the reference system is above (below) the stock obtained with the synthetic forecasting system. Figure 14 shows the spillage for the synthetic forecasting systems, expressed in terms of percentage of the spillage observed when using the reference system as inflow to the reservoir management model.





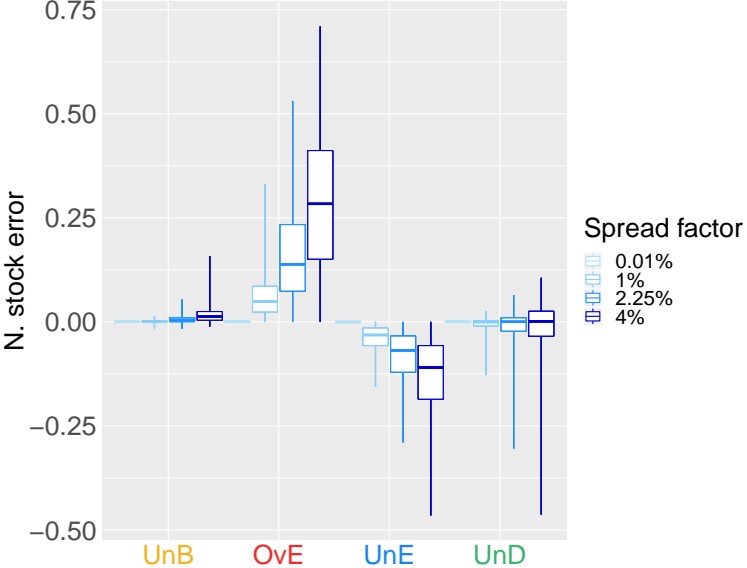

**Figure 13.** Normalized differences in stock between the operation with the reference system (i.e. the 'perfect system', where forecasts are equal to observations) and the operation with the synthetic forecasting systems: unbiased system (UnB), under-dispersed system (UnD), under-estimating system (UnE) and over-estimating system (OvE). Synthetic systems are based on ensembles generated with four different spread factors (0.01%, 1%, 2.25%, 4%). Boxplots represent the maximum value, percentiles 75, 50 and 25, and minimum value over 10 studied catchments.

We can see that the differences in spillage and stock are negligible for the unbiased system (UnB), although there is a tendency to have lower stock values than the reference system as the spread factor increases. This is explained by the slight overestimation of inflows that occurs when the spread factor is applied to this system. The same tendency to produce lower stock values than the reference system is observed for the synthetic system generated with an overestimation bias (OvE). A positive bias in a forecasting system leads to an excessive emptying of the reservoir (to make space for the incoming flows). Since the expected inflows are overestimated, the production schedule is more intense in order to handle the incoming volumes (i.e. more production hours as seen in Fig. 10). When the actual observed flows do not confirm the forecasts, but are lower, it becomes difficult to produce electricity at maximum capacity since there is a lack of water with regards to the forecasts. The production rate is then lowered, and we have more production hours with lower production rates, as indicated in Fig. 11. Also as a consequence of the systematic planning of emptying the reservoir, spillage is low and, in the configuration of our experiment, basically identical to the reference system (Fig. 14).

Differences in stock (Fig. 13) are negative for the biased systems that tend to underestimate the inflows overall (UnE) or the high inflows only (UnD). Underestimation leads to stock values more frequently higher than those that are obtained using a 'perfect' forecasting system. The reservoir management model takes into account the systematic low inflow forecasts throughout the study period, and the production planning tends to be minimized in order to maintain a stock in the reservoir





in the face of the low inflows that are forecast. This is why the total production of these synthetic systems is lower (Fig. 9).
However, the actual inflows to the reservoir, considered during the simulation phase of the management model, are greater than

those from the forecasts used to establish the management rule during the optimization phase. The volume of water entering
the reservoir is therefore higher than expected, resulting in an increase in the stock at each day of the study period, as shown
by the negative values in Fig. 13, notably for the UnE system. To face this, the number of production hours increases (Fig. 10),
but, given the unexpected higher inflows and the systematic forecast of low inflows, production can no longer be scheduled at
the best hours and at maximum capacity, resulting in more hours of production but at low capacity (Fig. 11 and Fig. 12, for the

UnE system). In summary, unexpected inflows have to be released through the turbines (production), but stock in the reservoir
has to be maintained high, due to the low inflow forecasts. Managing a reservoir while maintaining a high stock carries the risk
of poorly anticipating high flows, especially when the forecasting system has a recurrent bias towards underestimation, as in
the case of the UnE system. This also explains why the systems that tend to underestimate the inflows (UnE) or the high inflows
(UnD) have a higher amount of spilled water than the reference system (Fig. 14). These systems, which underestimate high

flows, have more difficulty managing flood events since they do not anticipate emptying the reservoir to create enough storage
capacity to store the high incoming volumes. The management model simulation phase follows the optimal command resulting
from the optimization phase. Since there is no re-optimization of excess water, the spilled water is not used for economic
purposes.



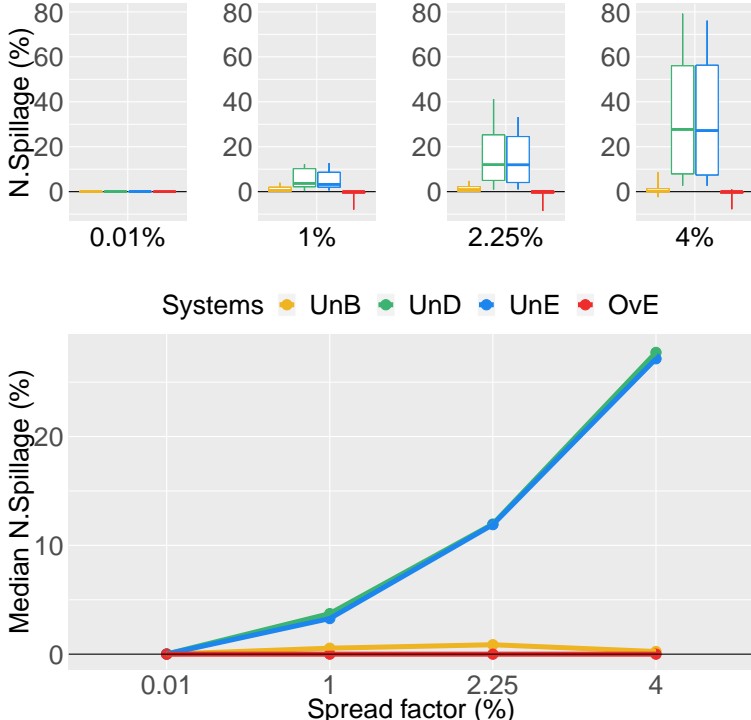

**Figure 14.** Spilled water from the management based on the synthetic ensemble forecasting systems as a percentage of the total spillage of the reference system (i.e. the 'perfect system', where forecasts are equal to observations): unbiased system (UnB, yellow), under-dispersed system (UnD, green), under-estimating system (UnE, blue) and over-estimating system (OvE, red). Synthetic systems are based on ensembles generated with four different spread factors (0.01%, 1%, 2.25%, 4%). Top: boxplots represent the maximum value, percentiles 75, 50 and 25, and minimum value over 10 studied catchments. Bottom: median value.

## 4  Conclusions

The overall purpose of this study was to better understand the link between forecast quality and forecast economic value in the case of the management of hydroelectric reservoirs. We investigated the impact of different forecast quality attributes of short-term (7-day ahead) hydrological forecasts on several output variables of a management reservoir model (revenue, production, production hours, stock values, spillage). Based on observed inflows, several synthetic forecasting systems were generated to mimic unbiased ensemble forecasts as well as biased forecasting systems of degraded quality in terms of reliability, sharpness

and accuracy. The synthetic forecast generator was developed and validated. It was shown that it can generate synthetic reliable ensemble forecasts, as well as ensemble forecasts with biases towards overestimation, underestimation and under-dispersion.

The ensemble mean of the different forecasting systems was then used as input to a reservoir management model, which was specifically built for the purposes of this study, based on a linear programming optimization algorithm. The optimization with 7-day ahead forecasts and the simulation with observed flows were carried out at the daily time step over a 4-year period,



from 2005 to 2008, at 10 catchment outlets in France. The management revenues were evaluated for each forecasting system based on the production and the electricity prices. Management results from the forecasting systems were compared with the results obtained from a reference system where forecasts are equal to observed flows.

This study showed that biased forecasts result in management revenues that are lower than the revenues of the unbiased forecasts or the reference system. Losses in revenue are stronger for forecasting systems that systematically overestimate reservoir inflows. In our configuration, positive bias (overestimation) leads to up to 3% of median economic losses, evaluated over the ten studied catchments and the 4-year period. Forecasting systems suffering from under-dispersion and under-estimation reach economic losses of about 1% and 1.5%, respectively. Nevertheless, due to the inefficiency of the management operations, these forecast systems result in large spillage, which can reach up to about 80% more than the spillage of the reference system in some catchments. This volume of water spilled (and not used for electricity production) was not taken into account in the assessment of the management revenue in this study, but it represents an extra economic loss for these systems. Although the percentages of economic losses obtained in this study are relatively small, they correspond to thousands to millions of euros per year of potential extra gains, according to the average electricity prices of the 2005-2008 period.

Overall, we can conclude that, given the configurations of the experiment of this study, and the tools used to generate forecasts of controlled quality and optimize the management of synthetic hydroelectric reservoirs located at actual catchment outlets in France, the quality of hydrological forecasts is clearly linked to their economic value in the hydropower sector: the worst (best) scores in forecast quality are associated with forecasting systems that display the lowest (higher) economic value. The analysis of the results showed that the quality of forecasts has an impact on management revenues, due to several factors:

- biased forecasting systems may result in more frequent hydroelectric production (i.e. more production hours), but the production is less often operated at higher capacities. Moreover, it occurs more often at lower electricity prices. Since management revenue is dependent on production capacity, this leads to lower management revenues. Optimal reservoir management is clearly inefficient in operating production within less hours but at higher capacity and when prices are higher.

- forecasting systems that present a positive bias result in a tendency of operations to keep the storage at lower levels so that the reservoir can be able to handle the high volumes expected. This impacts the optimal placement of production at the best hours (i.e. when prices are higher) and the opportunity to produce electricity at higher production rates. At the opposite, systems that suffer from underestimation biases tend to keep a high level of storage, which also influences the placement of production hours and the production rate, while also impacting the amount of spilled water (i.e. water lost and not used for electricity production). The amount and frequency of spillage may increase due to unexpected high flows incoming when the reservoir level is already too high and it is not possible to release enough water to create room to store the incoming high volumes.

- when using biased forecasting systems in hydropower reservoir management, production is not only planned during more hours at lower production rates but also at hours with lower median prices of electricity. This also impacts management revenues, since they also depend on the electricity price at the time of production.



This study showed the importance of the quality of hydrological forecasts in the management of a hydroelectric reservoir.
Measuring and removing forecast biases in hydrological forecasts, and more precisely overestimation biases, is therefore an
important step towards improving reservoir operations and potentially increasing hydropower revenues. This study was carried
out within a conceptual framework in order to address the challenges of quantitatively measuring the economic value of hydro-
meteorological forecasts in the hydropower sector and investigating the links between quality attributes and economic gains in
a controlled modelling environment. The modelling approaches adopted are certainly far from representing all the complexity
of hydropower management under uncertainties, which may include, for instance, taking into account actual sizes of existing
reservoirs, considering successions of over- and underestimation biases in the forecasts, and taking into account the hydraulic
head and more complex hydraulic management constraints. However, the level of schematization adopted in the experiments
carried out during this study proved to be adapted to obtaining the first orders of magnitude of the value of the forecast in
elementary situations.

This research work could have some main consequences. First, these orders of magnitude of the value of unbiased fore-
casts, as estimated at a first approximation, are sufficient to plead for devoted resources to continuously improve operational
hydrometeorological forecasting activities targeting the hydropower sector. Secondly, this first level of modelling, through the
proof-of-concept that it provides, is a step forward towards refined studies which, following a more detailed, long-term ap-
proach, may address the future challenges of the multi-use management of water resources. Multiple uses of water can also
benefit from better-informed hydropower operations. Finally, beyond the question of the value of the hydrometeorological
forecasts for the hydropower sector, the modelling approach of this study provides valuable support to test and demonstrate
also the decision process it simulates and how it may be impacted by forecasts of different quality.

*Data availability.* Streamflow data were provided by the French database "Banque HYDRO" and are available at http://www.hydro.eaufrance.
fr. Electricity prices data were produced by the EPEX-SPOT market and were provided by "Electricité de France". The synthetically gener-
ated hydrological forecasts and model outputs from the reservoir management model can be made available upon request.

*Author contributions.* All authors contributed to designing the experiment. The heuristics of the reservoir management model was originally
designed by IZ, MHR, RG and JG, and coded by IZ and MHR, with later additions from MC. The linear programming version used in this
study was coded by TO, in collaboration with MC. MC coded the generation of synthetic ensembles and carried out all the experiments.
MHR and GT supervised the study. RG and JG contributed to the hydropower applications. MC prepared the paper with contributions from
all co-authors.

*Competing interests.* The authors declare that they have no conflicts of interest.





*Acknowledgements.* This research was supported by the European Union's Horizon 2020 Research and Innovation programme under grant agreement N°641811 (IMPREX project).



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
