# Peer review of "Impact of the quality of hydrological forecasts on the management and revenue of hydroelectric reservoirs - a conceptual approach"

_Hydrology and Earth System Sciences, 2020_

## Referee Comment (RC1) · Anonymous Referee #1 · 13 Sep 2020

This is a review of "Impact of the quality of hydrological forecasts on the management and revenue of hydroelectric reservoirs – a conceptual approach" (please not there is a space missing between "management" and "and" in the title in the Copernicus manuscript submission system) by Cassagnole et al.

The manuscript presents an analysis of ensemble streamflow forecast quality and how it relates to hydropower value in a rolling horizon test bench, using controlled mechanisms to induce biases and under-dispersion over an unbiased reference. The ensemble forecasts that are generated are modified to be overestimated, underestimated and under-dispersed. The impacts of the ensemble spread are also investigated. The

analysis is performed on 10 hydropower systems in France, using historical energy process to estimate gains/losses in revenue of having forecast biases compared to a perfect forecast. The authors find that the unbiased forecast performs the best in terms of quality and value, and that the overestimated forecasts (positive bias) perform the worst, especially at high dispersion levels.

I thoroughly enjoyed reading this paper as it is presented in a very clear manner and well structured, and is a good foray into the little-explored world of forecast value in hydropower systems. I found the analysis to be robust and sound, with the literature review being both necessary and complete. I am particularly fond of the method to generate streamflow forecasts using a statistical distribution that is reshuffled into a reasonable temporal pattern using an ECC-based method.

After reading the manuscript a few times, I have highlighted a few points that I think are worth investigating as they could change the interpretation of some of the results for some users. Therefore I think that these few points I suggest below would make the paper even more solid and allow users from a broader audience to find interest in the study results. I recommend minor revisions as the changes are mostly all to the text and I don't think more simulations or analyses are required.

Specific comments: 1- Line 205-206: Here it is written that three additional values of D were used, which corresponds to the spread factors of 0.01%, 1%, 2.25% and 4% (4 values). It should be integrated with the previous sentence or explained better that the first value of 0.01% is from the D value of 0.01.

2- Figure 2: The forecasts for the UnE and the UnD are eerily similar, which is explained later in the discussion for other aspects but the discussion should refer to figure 2 to show how the forecasts are similar by reducing the high peaks (Figure 2 is not discussed after line 219). Making this link near lines 366-370, lines 379-382, lines 443-445, etc.

3- Lines 220-226: Could the LP solver be cited? Is it a commercial software such as

XPRESS, CPLEX, GUROBI, etc.? or is it in-house? Perhaps open source?

4- Line 239-240: Here, do the authors mean that the ensemble mean of the streamflow is fed to the solver (I think this is the case)? Or is the solver run on all members and the average decision taken? The latter could be interesting also to consider non-linearities in the ensemble forecasts. If the former is true, which I think it is, then I wonder why bother to generate ensemble streamflow forecasts at all? Why not simply generate deterministic forecasts with the desired properties (slightly biased, over/under -stimated, etc.?) under-dispersion, if it is symmetric, would be the same as the unbiased system, as seen in many figures. I understand it is not necessarily symmetric but the method employed (skewness of the log-normal distribution) could be corrected by another distribution... Basically, I think the authors should justify the use of an ensemble-based methodology if the solver only uses the mean value and the rolling-horizons test-bed only uses the ensemble average as well. Any difference between the unbiased and under-dispersed values would be due to the generation of artificial streamflow. If the authors used real-world streamflows (or simulations of streamflow using observed/forecast weather) then this point would not bother me as much...

5- Line 244-245: This means that there is no terminal water value, correct? i.e. the system would want to empty the reservoir at the end of the period to maximize benefits IF there was no constraint on drawdown volumes equal to the expected inflows? Perhaps the authors could comment on the impacts of this, as there would be no consideration of marginal head gains.

6- The fact that the optimization method is deterministic should introduce a deterministic bias, by which the optimization method does not account for uncertainty and thus is over-confident that it can maintain high-head without spilling (Philbrick and Kitandis 1999). Therefore slightly increasing the bias "tricks" the model into thinking there will be more water, forcing it to produce more energy and thus counteracting it's over-confidence on high water levels. In a real-world scenario, this would be observed as the reservoir head would increase efficiency and entice the optimization algorithm to

maximize revenues this way. But the setup in this study uses constant efficiency (line 252) with no change caused by reservoir head, making it much less representative of an actual system. The optimization has no need to optimize water levels, just make sure the reservoir doesn't crash or overtop. I think the authors need to add a section to the discussion to highlight these differences, because the way the abstract, discussion and conclusion are written, it could seem like the authors are saying that overestimation of forecasts is the worst possible solution, whereas in the real world it is probably the best option if using a deterministic optimization algorithm to counteract the optimization deterministic bias. I think the presented results would be much more in line with what would be seen if the optimization algorithm where stochastic (SDP or in the same vein), as the uncertainty is inherently included, which is not the case for deterministic solvers. [Philbrick, C. R. and Kitandis, P. K.: Limitations of Deterministic Optimization Applied to Reservoir Operations, J. Water Res. Pl.-ASCE, 125, 135–142, https://doi.org/10.1061/(ASCE)0733-9496(1999)125:3(135), 1999.] 

7- For the soft constraints, is there a penalty term in the cost function? Or is it considered as a hard constraint during the solving and then dealt with during the simulation part? If there is a penalty, could the information be provided?

8- Could the authors perhaps give an example (figure?) of the impact of increase in dispersion/spread using the D2 factor, compared to the effect of the under-dispersion? How does this affect the ensemble mean? Perhaps an example with a random date would help understand these impacts and differences (as the increase in spread is counteracted by the process of under-dispersion).

9- Figure 13: I think this figure could be changed to a 2x2 panel figure as in the previous figures, to keep things simpler and cleaner.

Typo/precisions needed: 1- Line 17: "…approximately 3% to 1% (in M€…" Here, I suppose the authors mean that the value of 1-3% represents Millions of Euros, however the way I read it is as if the units were M€ which would be moot as the difference is
relative. I suggest writing it as "which represents millions of Euros"

2- Line 28: squared kilometers → square kilometers

3- Line 281: non-respect → violation

4- Lines 487-492: These few sentences were quite confusing to read. I think they are technically correct, but reading them and parsing the information was somewhat difficult. Perhaps separating into a few more sentences and clarifying? Especially for the last sentence (lines 490-492).

---

## Referee Comment (RC2) · Anonymous Referee #2 · 14 Oct 2020

The paper studies the interactions of hydraulic forecasts with the economic value of the forecast. They go beyond the existing literature in the level detail that they investigate these interactions - studying how different biases in forecasts can interact with economic value (a question of both scientific but also practical importance). The experiments are performed on multiple catchments (10), for multiple types of forecasting biases (4), assessed by multiple measures of forecast quality (3), and multiple measures of forecast value are compared (5) - thus, I consider these to be very robust results.

Beyond this the authors demonstrate a good command of the literature, written English

of a high standard, and clear results with well supported conclusions.

In general, I would say just a little more effort might be made to help the reader understand where this work falls in terms of a practical perspective. The comments below will go some of the way to achieving this.

As far as I understood, the synthetic forecasts are not compared against historic forecasts that have been made nor against a real forecasting system. I understand this comparison would be outside the scope of the paper. However, have any of the cited authors who work with synthetic forecasts made this comparison? It seems to me that understanding whether the biases in synthetic forecasts reflect the types of biases in real forecasts is important information for a reader to know regarding the practical value of synthetic forecast-based studies.

L234/5: This seems a rather arbitrary choice for reservoir storage capacity. In fact, I am left slightly confused as to whether the reservoirs under study are real or not (I think they are not, but '10 reservoirs in France' is used in the abstract). If they are not, that needs to be explained more clearly. If they are real, then surely real capacities can be identified?

I was not clear from the explanation in 2.3, is there some accounting for the state of the reservoir storages at the end of the 7-day optimization period? It is OK if there is not, but the authors should note this, as it may lead to reservoirs becoming overdrawn in the long run.

Editorial comments:

L35/36: The word 'interesting' is an unusual choice, I would suggest 'beneficial'

L38: Grammar should be '..within integrated river basin management..'

L40: You can't have the 'most' optimal. Simply use the word 'optimal', or if you want to avoid the implications that releases are truly optimal, then use the word 'best'.

L44: Personally I would extend 'linear programming' to 'linear and nonlinear programming' since there are many nonlinear approaches (see any of the cited reviews for references) here that could not be said to fall under the term dynamic/heuristic programming.

Figures 11 & 12 - why switch to number of hours instead of % difference as in the other plots?

---

## Author Comment (AC1) · 27 Nov 2020

**Reply to the reviewer comments RC1: 'referee comments', by Anonymous Referee #1**

We would like to thank the Anonymous Referee #1 for the detailed review of our paper and the many constructive comments. In the following, we answered each comment individually. The reviewer comments are printed in gray italic font and our replies, in black.

**Specific comments:**

*1- Line 205-206: Here it is written that three additional values of D were used, which corresponds to the spread factors of 0.01%, 1%, 2.25% and 4% (4 values). It should be integrated with the previous sentence or explained better that the first value of 0.01% is from the D value of 0.01.*

Reply: Thank you for the remark. We will clarify this in the revised version.

*2- Figure 2: The forecasts for the UnE and the UnD are eerily similar, which is explained later in the discussion for other aspects but the discussion should refer to figure 2 to show how the forecasts are similar by reducing the high peaks (Figure 2 is not discussed after line 219). Making this link near lines 366-370, lines 379-382, lines 443-445, etc.*

Reply: We agree with the reviewer and we will modify accordingly in the revised version.

*3- Lines 220-226: Could the LP solver be cited? Is it a commercial software such as XPRESS, CPLEX, GUROBI, etc.? or is it in-house? Perhaps open source?*

Reply: The linear problem is solved by a third party library COIN Clp, which applies the simplex algorithm. The COIN Clp solver is called using a Python interface, called PuLP. We will add this information in the revised version.

*4- Line 239-240: Here, do the authors mean that the ensemble mean of the streamflow is fed to the solver (I think this is the case)? Or is the solver run on all members and the average decision taken? The latter could be interesting also to consider non-linearities in the ensemble forecasts. If the former is true, which I think it is, then I wonder why bother to generate ensemble streamflow forecasts at all? Why not simply generate deterministic forecasts with the desired properties (slightly biased, over/under -stimated, etc.?) under-dispersion, if it is symmetric, would be the same as the unbiased system, as seen in many figures. I understand it is not necessarily symmetric but the method employed (skewness of the log-normal distribution) could be corrected by another distribution: Basically, I think the authors should justify the use of an ensemblebased methodology if the solver only uses the mean value and the rolling-horizons test-bed only uses the ensemble average as well. Any difference between the unbiased and under-dispersed values would be due to the generation of artificial streamflow. If the authors used real-world streamflows (or simulations of streamflow using observed/forecast weather) then this point would not bother me as much:*

Reply: The solver is run with the ensemble mean in this paper. We used an ensemble of generated streamflows to approach the (real-world) ensemble operational forecasts used to recover the temporal evolution (ECC approach) and to retrieve ensemble characteristics (such as the under-dispersion generated with the UnD system) that are sometimes found in practice (as shown in Fig. 4). It should also be noted that we also investigated running the solver with each ensemble member (optimization step) and then averaging the decisions. We agree with the reviewer that this is also an interesting question to investigate. This is however the topic of another paper under preparation, where we used operational forecasts of different quality (with/without meteorological and/or hydrological post-processing) to feed the solver. With operational forecasts, we were also able to investigate the influence of the evolution of the quality of the forecasts with lead time, which we do not investigate in this paper since the quality (the bias) is identical at each lead time. This paper

would be too long if we had also included these aspects. Therefore, we opted to separate the studies: while this paper focuses on the influence of systematic bias on economic value, our paper under preparation focuses on the influence of the 'mean versus all members' approach on the output of the reservoir management model.

*5- Line 244-245: This means that there is no terminal water value, correct? i.e. the system would want to empty the reservoir at the end of the period to maximize benefits IF there was no constraint on drawdown volumes equal to the expected inflows? Perhaps the authors could comment on the impacts of this, as there would be no consideration of marginal head gains.*

Reply: The reviewer understands it correctly; there is no the terminal water value. As also noted by Reviewer#2, if there is no accounting for the state of the reservoir storage at the end of the period, the management model would normally want to empty the reservoir at the end of the period to maximize benefits. In fact, in our study, we implemented a weekly production as a 'soft constraint', which should not be higher than the weekly volume of water entering the reservoir. This prevents the reservoir from being emptied. This is an important issue pointed out by both reviewers and we will make it clear in the revised version.

*6- The fact that the optimization method is deterministic should introduce a deterministic bias, by which the optimization method does not account for uncertainty and thus is over-confident that it can maintain high-head without spilling (Philbrick and Kitandis 1999). Therefore slightly increasing the bias "tricks" the model into thinking there will be more water, forcing it to produce more energy and thus counteracting it's overconfidence on high water levels. In a real-world scenario, this would be observed as the reservoir head would increase efficiency and entice the optimization algorithm to maximize revenues this way. But the setup in this study uses constant efficiency (line 252) with no change caused by reservoir head, making it much less representative of an actual system. The optimization has no need to optimize water levels, just make sure the reservoir doesn't crash or overtop. I think the authors need to add a section to the discussion to highlight these differences, because the way the abstract, discussion and conclusion are written, it could seem like the authors are saying that overestimation of forecasts is the worst possible solution, whereas in the real world it is probably the best option if using a deterministic optimization algorithm to counteract the optimization deterministic bias. I think the presented results would be much more in line with what would be seen if the optimization algorithm where stochastic (SDP or in the same vein), as the uncertainty is inherently included, which is not the case for deterministic solvers. [Philbrick, C. R. and Kitandis, P. K.: Limitations of Deterministic Optimization Applied to Reservoir Operations, J. Water Res. Pl.-ASCE, 125, 135–142, https://doi.org/10.1061/(ASCE)0733-9496(1999)125:3(135), 1999.]*

Reply: This is a very relevant remark and we thank the reviewer for drawing attention to it. We will include a discussion on this issue in the revised version.

*7- For the soft constraints, is there a penalty term in the cost function? Or is it considered as a hard constraint during the solving and then dealt with during the simulation part? If there is a penalty, could the information be provided?*

Reply: There are penalty terms in the cost function associated with spills (maximum volume) and minimum volumes. They only intervene in the objective function of the LP, and not in the simulation part or the economic valuation (evaluation of gains). Penalties are based on the order of magnitude of the gains per hm3 (taking the maximum electricity price into account). The minimum volume penalty is calculated to always be greater than the potential gains, and the spill penalty 10 times the minimum volume penalty. We thank the reviewer for pointing this out and will provide the information in the revised version.

*8- Could the authors perhaps give an example (figure?) of the impact of increase in dispersion/spread using the D2 factor, compared to the effect of the under-dispersion? How does this affect the ensemble mean? Perhaps an example with a random date would help understand these impacts and differences (as the increase in spread is counteracted by the process of under-dispersion).*

Reply: The figure below represents the evolution of the mean of the synthetic ensemble forecast for the under-dispersed forecasting system and for the different spread factors (fig. 1R). We can see that the stronger the spread factor is, the further the mean of the ensemble will be from the observed value. In other words, the higher the spread factor, the more the high flows will be underestimated and the low flows overestimated.

[Figure]

*Figure 1R: Evolution of the mean of the synthetic ensemble forecast for the under-dispersed forecasting system and the different spread factors. Each color represents a spread factor value, ranging from light blue (spread factor = 0.01%) to dark blue (spread factor = 4%). The red line represents the observed flows.*

*9- Figure 13: I think this figure could be changed to a 2x2 panel figure as in the previous figures, to keep things simpler and cleaner.*

Reply: We thank the reviewer for the suggestion. We will modify the revised version with the figure below (caption remains the same):

[Figure]

**Typo/precisions needed:**

*1- Line 17: "approximately 3% to 1% (in MC)" Here, I suppose the authors mean that the value of 1-3% represents Millions of Euros, however the way I read it is as if the units were MC which would be moot as the difference is relative. I suggest writing it as "which represents millions of Euros"*

We will consider the suggestion in the revised version.

*2- Line 28: squared kilometers ! square kilometers*

We will correct it in the revised version.

*3- Line 281: non-respect ! violation*

We will correct it in the revised version.

*4- Lines 487-492: These few sentences were quite confusing to read. I think they are technically correct, but reading them and parsing the information was somewhat difficult. Perhaps separating into a few more sentences and clarifying? Especially for the last sentence (lines 490-492).*

We will consider the suggestion and clarify it in the revised version.

---

## Author Comment (AC2) · 27 Nov 2020

**Reply to the reviewer comments RC2: 'referee comments', by Anonymous Referee #2**

We would like to thank the Anonymous Referee #2 for the detailed review of our paper and the constructive comments. In the following, we answered each comment individually. The reviewer comments are printed in grey italic font and our replies, in black font.

Specific comments:

*1- In general, I would say just a little more effort might be made to help the reader under-stand where this work falls in terms of a practical perspective. The comments below will go some of the way to achieving this.*
*As far as I understood, the synthetic forecasts are not compared against historic forecasts that have been made nor against a real forecasting system. I understand this comparison would be outside the scope of the paper. However, have any of the cited authors who work with synthetic forecasts made this comparison? It seems to me that understanding whether the biases in synthetic forecasts reflect the types of biases in real forecasts is important information for a reader to know regarding the practical value of synthetic forecast-based studies.*

Reply: The reviewer understood it correctly. The synthetic forecasts are compared against observed flows to verify the technique used to create different biases in the forecasts. Forecasts from a real forecasting system are only used to apply the ECC method to retrieve temporal correlation, as presented in lines 179-188. These forecasts are based on operational models. We have not evaluated their quality in this paper (as the reviewer mentioned, it is out of the scope). However, as reference, the reader can find scores of forecast evaluation for a similar dataset in the paper by Zalachori et al. (2012), which focused on evaluating the impact of bias correction techniques applied to precipitation and to streamflow forecasts. This study shows that raw (without bias correction) operational streamflow ensemble forecasts display biases and under-dispersion, notably for the shorter lead times, with mean normalized RMSE values ranging from 1.7 to 2.4. These values are of the order of magnitude of the over-estimation biases introduced in our synthetic forecasts (Fig. 6, top).

Concerning the link between the quality of synthetically generated forecasts and the quality of forecasts used in practice in the studies we cited in our paper, we note the following:

- Lamontagne and Stedinger (2018) do emphasize that "Regardless of how they are generated, synthetic forecasts should replicate the important statistical properties of the real forecasts or the specified properties of a potential forecast product", referring to statistical properties in terms of mean, variance and accuracy. This is the basis for the conceptualization of their forecast generation approach. Their application is however illustrative. It supposes that a given forecast product has a given R2 and then generates synthetic time series of flows based on this information.

- Maurer and Lettenmaier (2004) worked with a measure of predictability developed in their previous studies as indication of potential seasonal predictability (fractional runoff variance explained by given predictors). They added errors to observed flows that are normally distributed with a mean of zero and a variance that is a function of this measure of predictability. Therefore, the comparison against practice relies on the fact that they use a range of values for this measure of predictability that are reported by the forecast skill evaluation study presented in Maurer and Lettenmaier (2003).

- Arsenault and Côté (2019) used a proxy for observed streamflow derived from the hydrological model initialized with empty reservoirs and driven by observed climate data (ESP-type forecasts). They then add bias to the ensemble means by multiplying the ESP forecast members by a factor that allows to shift the distribution upwards (factor >1) or downwards (factor <1). The factors chosen

resulted in bias that ranged from −7 % to +7 %. As for the relation to biases seen in operational practice, the authors only mention that: "Larger values were excluded because they were not necessary for exploring the behaviour of biases on the hydropower system operation".

We fully agree with the reviewer that understanding how much the biases in the synthetic forecasts reflect the actual biases in operational forecasts is important regarding the practical value of the conclusions drawn from studies based on synthetic forecasts. It should also be noted that, in operational forecasts, bias may vary according to the time of the year, the physical phenomenon causing high or low flows, and the catchment under consideration. Additionally, biases are often dependent on lead time, which is not the case in our synthetic forecasts. In the same context of the work we present in Zalachori et al. (2012), we observed that in some catchments the hydrometeorological forecasts (forecasts issued in the period 2005-2008) tend to under-estimate the flows, while in other catchments, the tendency is towards over-estimation (PhD Thesis I. Zalachori, 2013). The fact that we introduce the same bias over all time steps and catchments in the synthetic forecasts of our study makes it challenging to compare the synthetic forecasts with actual forecasts. Despite these limitations, we note that we have created the scenario of under-dispersed ensembles in our synthetic generation, where high flows are underestimated and low flows are overestimated (UnD synthetic forecasting system), to capture a feature that is often observed in operational hydrological forecasts due to the difficulties to capture peak flows in extreme events and to model highly-influenced low flows. We will make this important point clearer in the revised version of the paper.

*Zalachori, I., Ramos, M.H., Garçon, R., Mathevet, T., Gailhard, J., 2012. Statistical processing of forecasts for hydrological ensemble prediction: a comparative study of different bias correction strategies.* Advances in Science & Research*, vol. 8, p. 135 – 141.* [doi:10.5194/asr-8-135-2012](doi:10.5194/asr-8-135-2012))

*Zalachori, I., 2013. Prévisions hydrologiques d'ensemble : développements pour améliorer la qualité des prévisions et estimer leur utilité. Thèse de Doctorat, Irstea (Antony), AgroParisTech (Paris), 398 pp. [In French]*

*Maurer, E. P., and D. P. Lettenmaier, 2003: Predictability of seasonal runoff in the Mississippi River basin. J. Geophys. Res., 108, 8607, doi:10.1029/2002JD002555*

*2-L234/5: This seems a rather arbitrary choice for reservoir storage capacity. In fact, I am left slightly confused as to whether the reservoirs under study are real or not (I think they are not, but '10 reservoirs in France' is used in the abstract). If they are not, that needs to be explained more clearly. If they are real, then surely real capacities can be identified?*

Reply: The reviewer again understood it correctly. The reservoirs are not real reservoirs. We use actual observed streamflows of the 10 catchments that provide the inflows to the reservoirs, but we do not use the actual reservoir dimensions and operational characteristics. This will be explained more clearly in the revised version.

*3-I was not clear from the explanation in 2.3, is there some accounting for the state of the reservoir storages at the end of the 7-day optimization period? It is OK if there is not, but the authors should note this, as it may lead to reservoirs becoming overdrawn in the long run.*

Reply: It is well noted that if there is no accounting for the state of the reservoir storage at the end of the period, the management model would want to empty the reservoir at the end of the period to maximize benefits. In this study, we do not use water value (future benefits) to constraint the storage level at the end of the 7-day, but instead we implemented a weekly production as a 'soft constraint', which should not be higher than the weekly volume of water entering the reservoir. This prevents the reservoir from being emptied. This is an important issue also pointed out by Reviewer#1 and we will make it clear in the revised version.

Editorial comments:

*L35/36: The word 'interesting' is an unusual choice, I would suggest 'beneficial'*

We will modify it in the revised version.

*L38: Grammar should be '..within integrated river basin management..'*

We will correct it in the revised version.

*L40: You can't have the 'most' optimal. Simply use the word 'optimal', or if you want to avoid the implications that releases are truly optimal, then use the word 'best'.*

We will correct it to 'optimal' in the revised version.

*L44: Personally I would extend 'linear programming' to 'linear and nonlinear programming' since there are many nonlinear approaches (see any of the cited reviews for references) here that could not be said to fall under the term dynamic/heuristic programming.*

We will modify it in the revised version.

*Figures 11 & 12 - why switch to number of hours instead of % difference as in the other plots?*

The aim is to give also the order of magnitude of the differences.

---

## Author Response (AR1)

**Reply to the reviewers R1 & R2 comments**

**RC1: 'referee #1 comments'**

We would like to thank the Anonymous Referees #1 and #2 for their detailed reviews of our paper and the many constructive comments. We believe they improved our manuscript and we hope we made clearer the main aspects that were highlighted in the reviews. In the following, we answered each comment individually. The reviewer comments are printed in gray italic font and our replies, in black. The revised version marks in blue the parts of the text that were changed with regards to the original submission.

Specific comments:

*1- Line 205-206: Here it is written that three additional values of D were used, which corresponds to the spread factors of 0.01%, 1%, 2.25% and 4% (4 values). It should be integrated with the previous sentence or explained better that the first value of 0.01% is from the D value of 0.01.*

Reply: We added the following in Section 2.2 (step 2): "An ensemble forecast very close to the observed streamflows is generated from the synthetic ensemble forecasting model with a D coefficient equal to 0.01, corresponding to a spread factor of 0.01%. We then generated additional ensembles with D coefficients equal to 0.1, 0.15 and 0.2, which corresponds to spread factors of 1%, 2.25% and 4%."

*2- Figure 2: The forecasts for the UnE and the UnD are eerily similar, which is explained later in the discussion for other aspects but the discussion should refer to figure 2 to show how the forecasts are similar by reducing the high peaks (Figure 2 is not discussed after line 219). Making this link near lines 366-370, lines 379-382, lines 443-445, etc.*

Reply: We modified the revised version in Section 3.1 to refer to Fig. 2 as follows: "The resemblance between the two synthetic systems is also illustrated in Fig. 2.", when commenting Fig. 5, and later "reflecting their similarity when high peak flows are reduced (see Fig. 2).", when commenting Fig. 6.

*3- Lines 220-226: Could the LP solver be cited? Is it a commercial software such as XPRESS, CPLEX, GUROBI, etc.? or is it in-house? Perhaps open source?*

Reply: We added this information in section 2.3: "The linear problem is solved by the open-source solver COIN Clp, which applies the simplex algorithm. The solver is called using PuLP, a Python library for linear optimization."

*4- Line 239-240: Here, do the authors mean that the ensemble mean of the streamflow is fed to the solver (I think this is the case)? Or is the solver run on all members and the average decision taken? The latter could be interesting also to consider non-linearities in the ensemble forecasts. If the former is true, which I think it is, then I wonder why bother to generate ensemble streamflow forecasts at all? Why not simply generate deterministic forecasts with the desired properties (slightly biased, over/under -estimated, etc.?) under-dispersion, if it is symmetric, would be the same as the unbiased system, as seen in many figures. I understand it is not necessarily symmetric but the method employed (skewness of the log-normal distribution) could be corrected by another distribution: Basically, I think the authors should justify the use of an ensemble based methodology if the solver only uses the mean value and the rolling-horizons test-bed only uses the ensemble average as well. Any difference between the unbiased and under-dispersed values would be due to the generation of artificial streamflow. If the authors used real-world streamflows (or simulations of streamflow using observed/forecast weather) then this point would not bother me as much:*

Reply: The solver is run with the ensemble mean in this paper. We used an ensemble of generated streamflows to approach the (real-world) ensemble operational forecasts used to recover the temporal evolution (ECC approach) and to retrieve ensemble characteristics (such as the under-dispersion generated with the UnD system) that are sometimes found in practice (as shown in Fig. 4). It should also be noted that we also investigated running the solver with each ensemble member (optimization step) and then averaging the decisions. We agree with the reviewer that this is also an interesting question to investigate. This is however the topic of another paper under preparation, where we used operational forecasts of different quality (with/without meteorological and/or hydrological post-processing) to feed the solver. With operational forecasts, we were also able to investigate the influence of the evolution of the quality of the forecasts with lead time, which we do not investigate in this paper since the quality (the bias) is identical at each lead time. This paper would be too long if we had also included these aspects. Therefore, we opted to separate the studies: while this paper focuses on the influence of systematic bias on economic value, our paper under preparation focuses on the influence of the 'mean versus all members' approach on the output of the reservoir management model.

In the revised version, we added this in Section 2.3: "For each synthetic ensemble forecast, the reservoir management model is run with the mean of the members of the ensemble. Running the solver with each ensemble member and taking an average decision would also be possible, but would require additional investigation on the influence of extreme values of individual members on the decision, which is beyond the scope of this paper."

*5- Line 244-245: This means that there is no terminal water value, correct? i.e. the system would want to empty the reservoir at the end of the period to maximize benefits IF there was no constraint on drawdown volumes equal to the expected inflows? Perhaps the authors could comment on the impacts of this, as there would be no consideration of marginal head gains.*

Reply: The reviewer understands it correctly; there is no the terminal water value. As also noted by Reviewer#2, if there is no accounting for the state of the reservoir storage at the end of the period, the management model would normally want to empty the reservoir at the end of the period to maximize benefits. In fact, in our study, we implemented a weekly production as a 'soft constraint', which should not be higher than the weekly volume of water entering the reservoir. This prevents the reservoir from being emptied. This is an important issue pointed out by both reviewers and we hope we made it clear in the revised version, where we added this in section 2.3: "We did not use final water values to account for the state of the reservoir storage at the end of the 7-day optimization period. However, we implemented the weekly production as a soft constraint, which should not be higher than the weekly volume of water entering the reservoir. This prevents the model to empty the reservoirs at the end of the period."

*6- The fact that the optimization method is deterministic should introduce a deterministic bias, by which the optimization method does not account for uncertainty and thus is over-confident that it can maintain high-head without spilling (Philbrick and Kitandis 1999). Therefore slightly increasing the bias "tricks" the model into thinking there will be more water, forcing it to produce more energy and thus counteracting it's overconfidence on high water levels. In a real-world scenario, this would be observed as the reservoir head would increase efficiency and entice the optimization algorithm to maximize revenues this way. But the setup in this study uses constant efficiency (line 252) with no change caused by reservoir head, making it much less representative of an actual system. The optimization has no need to optimize water levels, just make sure the reservoir doesn't crash or overtop. I think the authors need to add a section to the discussion to highlight these differences, because the way the abstract, discussion and conclusion are written, it could seem like the authors are saying that overestimation of forecasts is the worst possible solution, whereas in the real world it*

*is probably the best option if using a deterministic optimization algorithm to counteract the optimization deterministic bias. I think the presented results would be much more in line with what would be seen if the optimization algorithm where stochastic (SDP or in the same vein), as the uncertainty is inherently included, which is not the case for deterministic solvers. [Philbrick, C. R. and Kitandis, P. K.: Limitations of Deterministic Optimization Applied to Reservoir Operations, J. Water Res. Pl.-ASCE, 125, 135–142, https://doi.org/10.1061/(ASCE)0733-9496(1999)125:3(135), 1999.]*

Reply: This is a very relevant remark and we thank the reviewer for drawing attention to it. We included the following in the revised version (section 4): "In this study, we used a deterministic optimization model based on linear programming. It allowed us to set up the framework for the analysis of different sets of synthetic forecasts of different biases over a long time period and several locations with different climatic conditions. However, deterministic optimization methods also have drawbacks. They do not allow to account for inflow uncertainty, which limits the use of ensemble forecasts at its highest potential value, and the solution provided may not be optimal, especially under high-flow scenarios (Philbrick and Kitandis, 1999). In our study, this can have a particular impact on the more extreme conditions of the synthetic forecast system that overestimates the observed flows. With a constant efficiency of the power plant and no influence of the reservoir head, as considered in the configuration of our study, overestimations may misguide the model when it searches to maximize revenues. Further studies using the predictive distribution of the ensemble forecasts and stochastic optimization models are avenues that could be explored."

*7- For the soft constraints, is there a penalty term in the cost function? Or is it considered as a hard constraint during the solving and then dealt with during the simulation part? If there is a penalty, could the information be provided?*

Reply: There are penalty terms in the cost function associated with spills (maximum volume) and minimum volumes. We added the following information in the revised version (section 2.3): "There are penalty terms in the cost function associated with spills and minimum volumes, which intervene in the objective function during the optimization. Penalties are based on the order of magnitude of the gains per $hm^3$ (taking the maximum electricity price into account). The minimum volume penalty is calculated to always be greater than the potential gains, and the spill penalty is ten times the minimum volume penalty. For instance, for a gain of 8 per hm3, the order of magnitude will be 1; the power to 10, zero; the minimum volume penalty, 10; the spill penalty, 100."

*8- Could the authors perhaps give an example (figure?) of the impact of increase in dispersion/spread using the D2 factor, compared to the effect of the under-dispersion? How does this affect the ensemble mean? Perhaps an example with a random date would help understand these impacts and differences (as the increase in spread is counteracted by the process of under-dispersion).*

Reply: The figure below represents the evolution of the mean of the synthetic ensemble forecast for the under-dispersed forecasting system and for the different spread factors (fig. 1R). We can see that the stronger the spread factor is, the further the mean of the ensemble will be from the observed value. In other words, the higher the spread factor, the more the high flows will be underestimated and the low flows overestimated. For the sake of conciseness, we did not add this figure in the revised version.

[Figure]

*Figure 1R: Evolution of the mean of the synthetic ensemble forecast for the under-dispersed forecasting system and the different spread factors. Each color represents a spread factor value, ranging from light blue (spread factor = 0.01%) to dark blue (spread factor = 4%). The red line represents the observed flows.*

*9- Figure 13: I think this figure could be changed to a 2x2 panel figure as in the previous figures, to keep things simpler and cleaner.*

Reply: We modified the figure in the revised version:

[Figure]

Typo/precisions needed:

*1- Line 17: "approximately 3% to 1% (in MC)" Here, I suppose the authors mean that the value of 1-3% represents Millions of Euros, however the way I read it is as if the units were MC which would be moot as the difference is relative. I suggest writing it as "which represents millions of Euros"*
Corrected to the following: "Overall, the losses (which amount to millions of Euros) represent approximately 1% to 3% of the revenue over the study period".

*2- Line 28: squared kilometers ! square kilometers*
Corrected.

*3- Line 281: non-respect ! violation*
Corrected.

*4- Lines 487-492: These few sentences were quite confusing to read. I think they are technically correct, but reading them and parsing the information was somewhat difficult. Perhaps separating into a few more sentences and clarifying? Especially for the last sentence (lines 490-492).*
We broke it into several sentences and it reads as follows in the revised version: "These two biased forecasts tend to produce more than the reference system when the production rate is low. The reference system counts very few hours of production rate in classes C1, C2 and C3 of low production rate. Moreover, the biased systems tend to produce less when the production rate is closer to the maximum production capacity. The difference between the total production of the UnE system and that of the reference system (Fig. 9) is therefore explained by two factors: the high frequency of production of the UnE system in class C1 of low production rate, and its low frequency of production in class C4 (which is the one of highest production rate)."

**RC2: 'referee #2 comments'**

We would like to thank the Anonymous Referee #2 for the detailed review of our paper and the constructive comments. In the following, we answered each comment individually. The reviewer comments are printed in grey italic font and our replies, in black font.

Specific comments:

*1- In general, I would say just a little more effort might be made to help the reader under-stand where this work falls in terms of a practical perspective. The comments below will go some of the way to achieving this.*
*As far as I understood, the synthetic forecasts are not compared against historic forecasts that have been made nor against a real forecasting system. I understand this comparison would be outside the scope of the paper. However, have any of the cited authors who work with synthetic forecasts made this comparison? It seems to me that understanding whether the biases in synthetic forecasts reflect the types of biases in real forecasts is important information for a reader to know regarding the practical value of synthetic forecast-based studies.*

Reply: This is an interesting and relevant point indeed. We added the following in the revised version:

In Section 1: "For practical applications, the value of synthetic forecasts is related to how the biases in these forecasts reflect the actual biases encountered in operational forecasts. In operational hydrological forecasts, biases may vary according to the time of the year, the magnitude of flows (different biases may be observed for high and low flows), the catchment and its climatic conditions, among others. Additionally, forecast biases are often dependent on lead time. Lamontagne and Stedinger (2018) emphasized that synthetic forecasts should replicate the most important statistical properties (mean, variance, accuracy) of the real forecasts, or the specified properties of a potential forecast product to be analysed. Arsenault and Côté (2019) explained they excluded larger variations of biases in their study since this would not be of additional help in exploring the behaviour of biases on the hydropower system operation of their case study."

In Section 2.2: "The quality of these operational forecasts, for a similar catchment dataset and evaluation period, is discussed in Zalachori et al. (2012)"
At the end of Section 4: "Finally, we note that the range of NRMSE values of the OvE system comprises the range of normalized RMSE values found by Zalachori et al. (2012) (NRMSE ranging from 1.7 to 2.4) when analyzing the raw (without bias correction) operational forecasts over a similar dataset of catchments."

*2-L234/5: This seems a rather arbitrary choice for reservoir storage capacity. In fact, I am left slightly confused as to whether the reservoirs under study are real or not (I think they are not, but '10 reservoirs in France' is used in the abstract). If they are not, that needs to be explained more clearly. If they are real, then surely real capacities can be identified?*

Reply: The reviewer understood it correctly. In the revised version, the abstract now reads: "Flows from ten catchments in France are synthetically generated over a 4-year period to obtain forecasts of different quality in terms of accuracy and reliability. These forecasts define the inflows to ten hydroelectric reservoirs, which are conceptually parametrized." In Section 2.3, we have the following: "We do not use the actual reservoir dimensions and operational characteristics, although the inflows from the synthetic hydrological forecasts reflect the actual hydrological variability."

*3-I was not clear from the explanation in 2.3, is there some accounting for the state of the reservoir storages at the end of the 7-day optimization period? It is OK if there is not, but the authors should note this, as it may lead to reservoirs becoming overdrawn in the long run.*

Reply: There is no accounting for the water value or the state of the reservoir storage at the end of the period, but some penalty terms prevent the reservoir from emptying (see also our reply to Reviewer #1).
In the revised version, we added the following in Section 2.3: "There are penalty terms in the cost function associated with spills and minimum volumes, which intervene in the objective function during the optimization. Penalties are based on the order of magnitude of the gains per $hm^3$ (taking the maximum electricity price into account). The minimum volume penalty is calculated to always be greater than the potential gains, and the spill penalty is ten times the minimum volume penalty. For instance, for a gain of 8 per hm3, the order of magnitude will be 1; the power to 10, zero; the minimum volume penalty, 10; the spill penalty, 100.", and also: "We did not use final water values to account for the state of the reservoir storage at the end of the 7-day optimization period. However, we implemented the weekly production as a soft constraint, which should not be higher than the weekly volume of water entering the reservoir. This prevents the model to empty the reservoirs at the end of the period."

Editorial comments:

*L35/36: The word 'interesting' is an unusual choice, I would suggest 'beneficial'*
We modified it as suggested.

*L38: Grammar should be '..within integrated river basin management..'*
Corrected.

*L40: You can't have the 'most' optimal. Simply use the word 'optimal', or if you want to avoid the implications that releases are truly optimal, then use the word 'best'.*
We corrected to 'optimal' only.

*L44: Personally I would extend 'linear programming' to 'linear and nonlinear programming' since there are many nonlinear approaches (see any of the cited reviews for references) here that could not be said to fall under the term dynamic/heuristic programming.*
We agree and modified it as suggested.

*Figures 11 & 12 - why switch to number of hours instead of % difference as in the other plots?*
The aim is to give also the order of magnitude of the differences.